# Ruelle-Pollicott resonances of diffusive U(1)-invariant qubit circuits

**Urban Duh⋆ and Marko Žnidarič**

Physics Department, Faculty of Mathematics and Physics, University of Ljubljana, 1000 Ljubljana, Slovenia

⋆ urban.duh@fmf.uni-lj.si

## Abstract

We study Ruelle-Pollicott resonances of translationally invariant magnetization-conserving qubit circuits via the spectrum of the quasi-momentum-resolved truncated propagator of extensive observables. Diffusive transport of the conserved magnetization is reflected in the Gaussian quasi-momentum $k$ dependence of the leading eigenvalue (Ruelle-Pollicott resonance) of the truncated propagator for small $k$. This, in particular, allows us to extract the diffusion constant. For large $k$, the leading Ruelle-Pollicott resonance is not related to transport and governs the exponential decay of correlation functions. Additionally, we conjecture the existence of a continuum of eigenvalues below the leading diffusive resonance, which governs non-exponential decay, for instance, power-law hydrodynamic tails. We expect our conclusions to hold for generic systems with exactly one U(1) conserved quantity.

# 1 Introduction

One of the reasons why physics is so successful in describing nature is because oftentimes simple descriptions suffice. This is so because even simple rules, given for instance by a Hamiltonian, can lead to complicated behavior. How and which simple rules lead to complex behavior is studied within the theory of dynamical systems. Some systems (rules) lead to predictable regular dynamics, the extreme case being integrable systems, while others lead to complex behavior, the extreme case being chaotic systems for which predicting the future for a specific initial condition becomes impossible due to exponential complexity growth [1].

In many-body quantum systems, complexity can grow exponentially also with the system size. Such exponentially growing Hilbert space size is what allows for outperformance of quantum devices over classical, and indeed, one of the premier uses of present day digital quantum computers and analog simulators is to simulate quantum many-body systems [2]. An interesting question is: What new effects can arise from the combination of exponential complexity due to chaos and that from the many-body nature? To begin with, it is not entirely clear how to define or quantify quantum chaos in many-body quantum systems. One can try to use theoretically appealing concepts, like dynamical entropies, which are difficult to use in practice, or one can use a concept that is easy to study numerically, for instance one of the various spectral quantifiers [3]. Both have a drawback that in a many-body system they are hardly observable quantities.

One can, however, abandon theoretical elegance for practical relevance and try to directly study measurable objects. An example is the correlation function

$$C_{AB}(t) := \langle A(t)B(0) \rangle - \langle A(t) \rangle \langle B(0) \rangle , \qquad (1)$$

where $\langle \bullet \rangle$ denotes the average over an appropriate ensemble. A defining property of mixing systems is that correlation functions decay to zero, while for chaotic systems we expect more, namely, that $C_{AB}(t)$ decay exponentially at long enough times, $C_{AB}(t) \sim e^{-\nu t}$. Time-dependent correlation functions specifically allow us to study dynamics, and through linear response also get access to non-equilibrium properties. A prime example is the transport of globally conserved quantities, like energy, charge, or magnetization.

While many different methods have been proposed and employed to study transport in many-body quantum systems, for a review see Ref. [4], here we propose and test a new one that is based on the dynamics of operators. It directly works in the thermodynamic limit, i.e., in infinite-sized systems, with the only approximation being truncating the dynamics to operators with a finite local support size. It builds on a decades-old idea of Ruelle-Pollicott (RP) resonances [5–7] that we upgrade and adapt to translationally invariant many-body quantum circuits so that it allows for easy calculation of the transport dynamical exponent and phenomenological constants, for instance, the diffusion constant.

## 1.1 Ruelle-Pollicott resonances

The idea of RP resonances is to get the decay rate $\nu$ of the asymptotic exponential decay of correlation functions,

$$C_{AB}(t) \sim e^{-\nu t} = \lambda^t , \qquad (2)$$

where we introduced $\lambda := e^{-\nu}$. While specific observables can decay with their own rate, there is the slowest decay rate $\nu$, whose inverse is the system's relaxation time. It was shown in Refs. [5,6] that in a class of strongly chaotic classical systems $\lambda$ does not depend on the chosen observable, but is rather an intrinsic property of the system. In such cases, $\lambda$ is referred to as the (leading) RP resonance of the system. While one could get $\lambda$ by

simply calculating the appropriate correlation function, the idea is to get $\lambda$ directly from the generator of dynamics (after all, $\lambda$ is an inherent property of the system, not of the observables).

With classical systems in mind, the starting point is the propagator $\mathcal{U}$, either the Frobenius-Perron propagator of densities, or the Koopman propagator of observables [7,8]. $\mathcal{U}$ is unitary in the appropriate space (e.g., in the space of $L^2$ functions), its eigenvalues lying on the unit circle. One can, however, find poles in the analytic continuation of the resolvent $\mathcal{R}(z) := 1/(z - \mathcal{U})$ inside the unit circle. It was shown in specific systems [9] that they correspond exactly to RP resonances, the largest being the mentioned $\lambda$.

Performing the analytic continuation for generic systems is not feasible. A more practical approach, which can be carried out either analytically or numerically, is introducing some kind of dissipation or coarse-graining (i.e., a non-unitarity) in our system. This leads to a non-unitary propagator $\mathcal{U}(\varepsilon)$, where $\varepsilon$ measures the strength of the non-unitarity. The eigenvalues of $\mathcal{U}(\varepsilon)$ now lie inside the unit circle and, analogous to the RP resonances in the unitary case, govern the decay of generic correlation functions. It is conjectured that in the unitary limit $\varepsilon \to 0$ the naive expectation that the eigenvalues uniformly converge to the unit circle is not correct – they converge to a smaller radius inside the unit circle, which corresponds to an RP resonance. This procedure was used in classical systems, with the non-unitarity being either white noise [10] or a phase space coarse-graining procedure [11].

Less is known about RP resonances of quantum systems. Initially, they have been studied in the semi-classical regime [12–14]. More recently, weak Lindblad dissipation (an analog of noise in classical systems) has been suggested [15] as a method of extracting $\lambda$ via the Liouvillian gap in the limit of infinite system size and zero noise, with the results for a few specific solvable models [16–18] agreeing with $\lambda$ calculated directly.

We shall use a different path by considering the Heisenberg propagator of observables $\mathcal{U}(A) := U^\dagger A U$, where $U$ is the unitary propagator, truncating $\mathcal{U}$ to an appropriate operator subspace. In this setting, the RP resonance is expected to correspond to the leading eigenvalue of $\mathcal{U}$. The method was first used on translationally invariant local operators in Ref. [19] (which can be thought of as akin to a phase space coarse-graining) in the kicked Ising model [19–21], and more recently followed by additionally resolving the quasi-momentum [22]. We shall use this quasi-momentum-resolved truncated propagator. The truncated propagator can be very handy not just in studies of chaotic systems, but more broadly, for instance, in identifying unknown constants of motion in cellular automata [23,24], or discovering that all homogeneous U(1) conserving quantum circuits are integrable [25,26].

We note that the recently much discussed Krylov space method can be viewed as a particular way of truncation (in powers of time in the Taylor expansion, i.e., nested commutators) and can thus be related to RP resonances [27,28].

## 1.2   Summary of results

In this work we study RP resonances of systems with one conserved quantity. We focus on many-body quantum circuits with a local U(1) conserved quantity and local gates. Such systems, with their associated transport, are of direct experimental interest for NISQ machines [29–31], and are the simplest theoretical setting with a conserved quantity[1]. Our generic conclusions are demonstrated on magnetization-conserving circuits with 3-site interaction and 3-site translational invariance.

---

[1]Implementation of the truncated propagator method is simpler for quantum circuits compared to Hamiltonian systems due to a sharp light-cone. In addition, one has conserved magnetization with a 1-local density instead of conserved energy with a 2-local density.

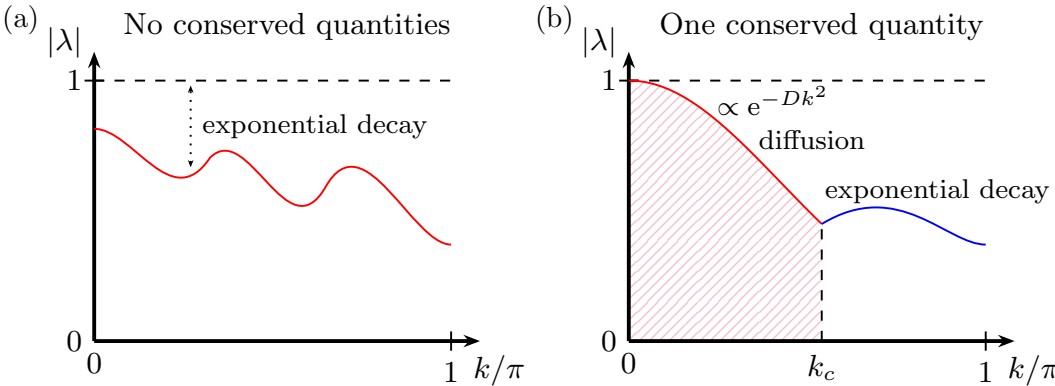

Figure 1: Diagram of the $k$-dependent RP resonance spectrum in homogeneous systems without (a), and with one conserved quantity (b). Labels in figure (b) show which parts of the spectrum are responsible for which physical behavior of the system. The shaded region in figure (b) denotes the location of conjectured continuums of RP resonances, which we cannot observe numerically with the methods used. We show only positive $k$ since $|\lambda(k)| = |\lambda(-k)|$, for details see Sec. 2.2.

As mentioned, we use the truncated operator propagator [19] in its quasi-momentum-resolved form [22], which allows us to extract transport properties from the momentum dependence of RP resonances. Some work has been done on the connection between RP resonances and diffusion in classical systems [32, 33], and conserved quantities have been briefly mentioned in previous Lindblad weak dissipation works [15, 18], as well as for the truncated propagator of the kicked Ising model in a prethermalized regime (or the Hamiltonian limit) in a recent preprint [34]. We also note that a somewhat related idea of dropping operators with many non-identity Pauli operators has been used in Ref. [35], where the authors introduce a numerical Lindblad evolution method that aims to reach the long-time hydrodynamic behavior.

We provide the first comprehensive study of the $k$-dependence of RP resonances, showing that (i) the quasi-momentum dependence of the leading eigenvalue of the truncated propagator and thus the leading RP resonance $\lambda_1(k)$ is in general non-trivial (Fig. 1a), even in a system without any conserved quantities where $|\lambda_1(k)|$ is gapped away from 1. For instance, the decay need not be always the slowest at $k = 0$, i.e., for translationally invariant observables. (ii) In systems with conserved magnetization, one expectedly has $\lambda_1(0) = 1$ with the corresponding eigenvector being the conserved magnetization, and, more interestingly, from $\lambda_1(k)$ for small $k$ one can extract the transport dynamical exponent, and, specifically in the case of diffusion, the value of the diffusion constant from $|\lambda_1(k)| = e^{-Dk^2}$ (Fig. 1b). At large $k$, the diffusive nature of the leading RP resonance ends due to an eigenvalue crossing at $k_c$, resulting in an exponential decay of generic observables with sufficiently large quasi-momentum $k$. (iii) We discuss and conjecture an RP continuum below $|\lambda_1(k)|$ (Fig. 1b) that is related to non-exponential decay of correlation functions and corrections to diffusive transport.

## 2 Truncated quasi-momentum-dependent propagator of extensive observables

In this section, we describe the quasi-momentum-dependent propagator of local observables, first introduced for $k = 0$ in Ref. [19] and extended to any $k$ in Ref. [22]. We are considering qubit circuits with propagators for 1 unit of time $U$ that have translational invariance by $s$ sites, and that can, due to a sharp light-cone, spread local operators supported on $r$ consecutive sites by at most $\delta r$ sites at each edge. For instance, a brickwall circuit composed of all the same gates (Fig. 2a) is invariant under translation by $s = 2$ sites and has $\delta r = 2$, while a 3-layered brickwall circuit with 3-qubit gates (Fig. 3a) has $s = 3$ and $\delta r = 6$. Previous implementations [19–22] were for the kicked Ising model that has $s = 1$ and $\delta r = 1$, or for the (2-layer) brickwall circuits [25, 26]. We generalize it to work for translationally invariant quantum circuits with any geometry, i.e., any $s$ and $\delta r$.

First, we express the Heisenberg propagator of observables

$$\mathcal{U}(A) := U^\dagger A U \tag{3}$$

in an operator basis of an infinite system where the quasi-momentum $k$ is a good quantum number. We then truncate (i.e., project) the observable $\mathcal{U}(A)$ back to observables with densities supported on at most $r$ sites, thus obtaining a non-unitary truncated propagator $\mathcal{U}^{(r)}$. In the limit of the non-unitarity going to zero, that is $r \to \infty$, we expect to extract RP resonances as the "frozen" eigenvalues of the truncated propagator $\mathcal{U}^{(r)}$.

### 2.1 Brief description

In order to study the Heisenberg propagator with good quasi-momentum $k$, we must first define a basis of operators with good $k$. Any observable in a finite system of $N$ lattice sites with good quasi-momentum $k$ can be written in the following way

$$A = \sum_{j=0}^{N/s-1} e^{-ikj} \mathcal{S}^{sj}(a), \tag{4}$$

where $\mathcal{S}$ is the 1-site translation super-operator to the right (i.e., $\mathcal{S}(a \otimes \mathbb{1}) = \mathbb{1} \otimes a$), $a$ is a local operator and the sum runs over all $j \in \{0, \ldots, N/s - 1\}$ possible translations by $s$ sites. In a finite system, $k$ takes values $\frac{2\pi s}{N} j$, however, at the end we will be interested in the thermodynamic limit $N \to \infty$[2], where we can take $k \in (-\pi, \pi]$. One can check that $A$ really is an eigenvector of the translation super-operator, $\mathcal{S}(A) = e^{ik} A$. Such $A$ will be referred to as an *extensive observable*.

We wish to work with local operators $a$ as proxies for extensive observables $A$. It is thus crucial to restrict ourselves to a space of local operators in which the correspondence between local operators and extensive observables is one-to-one. Namely, in Eq. (4), different $a$ can result in the same $A$. For example, both $a = \sigma_1^z$ and $a = \frac{1}{2}\left(\sigma_1^z + \sigma_{1+s}^z\right)$ represent the same $A$. Here $\sigma_j^\alpha$ denotes the $\alpha$ Pauli matrix acting on site $j$. To have a one-to-one correspondence, such translations of components of $a$ by $s$ sites must be prohibited. This can be done in various ways, we choose to write the extensive observable in the following form

$$A = \sum_{m=0}^{s-1} \sum_{j=0}^{N/s-1} e^{-ikj} \mathcal{S}^{sj+m}(a_m), \tag{5}$$

---

[2]Formally, RP resonances can only exist in systems with infinite-dimensional Hilbert spaces. In finite systems, correlation functions always converge to a nonzero constant.

where $\mathcal{S}^{sj+m}(a_m)$ can be thought of as operators starting on sites $sj + m$ (i.e., on all sites $i$, where $i \pmod{s} = m$). In other words, due to translational invariance by $s$ sites we have a local unit cell of $s$ sites, so that the sum over $m$ runs over sites within a unit cell, while $j$ runs over all unit cells. In this formulation it is now easy to see that we have a one-to-one correspondence between the set of $\{a_m\}_{m=0}^{s-1}$ and the extensive observable $A$ if we enforce that $a_m$ do not contain components acting trivially on the first site. Explicitly, $a_m$ must be in the space spanned by the set (basis) $\mathcal{P}^{(r)}$ of Pauli strings without identities acting on the first site:

$$p_0 := \{\mathbb{1}\} \cup p, \qquad \text{where} \qquad p := \{\sigma^x, \sigma^y, \sigma^z\}, \tag{6}$$
$$\mathcal{P}^{(r)} := \{p_1 \otimes p_2 \otimes \cdots \otimes p_r \mid p_1 \in p \ \wedge \ p_{j\neq 1} \in p_0\}.$$

In this notation, therefore,

$$a_m \in \mathrm{Span}\left(\mathcal{P}^{(r)}\right) \text{ for some } r. \tag{7}$$

Such $a_m$ will be referred to as *local densities*. Additionally, we define their support to be the smallest $r$ such that $a_m \in \mathrm{Span}\left(\mathcal{P}^{(r)}\right)$. Analogously, the support of an extensive observable $A$ is defined to be the smallest $r$, such that $a_m \in \mathrm{Span}\left(\mathcal{P}^{(r)}\right)$ for all $m \in \{0, \ldots, s-1\}$.

The basis of extensive observables can now be written down, one merely takes all possible local basis elements $\mathcal{P}^{(r)}$ on all possible starting positions $m$. The basis elements are therefore

$$B_k^{(m,b)} := \sum_{j=0}^{N/s-1} e^{-ikj} \mathcal{S}^{sj+m}(b), \tag{8}$$

where $b \in \mathcal{P}^{(r)}$ for some $r$. Moreover, $B_k^{(m,b)}$ for all $k, m$ and $b \in \mathcal{P}^{(r)}$ are an orthonormal basis of the space of extensive observables w.r.t. the *extensive* Hilbert-Schmidt inner product, $\langle\!\langle A, B \rangle\!\rangle := \frac{s}{N2^N} \mathrm{tr}\, A^\dagger B$. The number of basis elements of support $r$ (or less) is $\frac{3s}{4} 4^r$. Importantly, the identity operator $\mathbb{1}$ is excluded from this basis. This is convenient, since it is always a conserved quantity in unitary dynamics and is, therefore, not of interest in the present application. For details and derivations regarding the space of extensive observables see Appendix A.1.

It is now easy to calculate the matrix elements of the Heisenberg propagator in the basis of extensive observables by simply taking the inner product

$$[\mathcal{U}_k]_{(m,b),(m',b')} = \left\langle\!\left\langle B_k^{(m,b)}, \mathcal{U}\left(B_k^{(m',b')}\right) \right\rangle\!\right\rangle. \tag{9}$$

In this context, we call $\mathcal{U}$ the *propagator of extensive observables* (i.e., the Heisenberg propagator in a particular basis).

## 2.2 Truncation and Ruelle-Pollicott resonances

The Heisenberg propagator in Eq. (9) evaluated in the entire (infinite-dimensional) Hilbert space is a unitary operator. We wish to introduce a non-unitarity by restricting ourselves to some subspace. There are different options, one of the most natural ones is restricting ourselves to the space of support $r$ defined in Eq. (6). Projection of the Heisenberg propagator to this space results in the *truncated propagator of extensive observables* $\mathcal{U}_k^{(r)}$ with matrix elements

$$\left[\mathcal{U}_k^{(r)}\right]_{(m,b),(m',b')} = \sum_{j=-1}^{1} e^{ikj} \left\langle \mathcal{S}^{sj+m}(b), \mathcal{U}\left(\mathcal{S}^{m'}(b')\right) \right\rangle, \tag{10}$$

where $b, b' \in \mathscr{P}^{(r)}$ (6), $m, m' \in \{0, \ldots, s - 1\}$ and $k \in (-\pi, \pi]$. The RHS is expressed only in terms of local densities and the angled brackets denote the *local* Hilbert-Schmidt inner product, $\langle a, b \rangle := \frac{1}{2^N} \operatorname{tr} a^\dagger b$. Importantly, the truncated propagator is block-diagonal with blocks indexed by the quasi-momentum $k$. The written form is valid if $\delta r \leq s$. More generally, the range of values of $j$ over which one has to sum in Eq. (10) depends on the spreading $\delta r$ of the circuit. For the derivation and further details see Appendix A.2.

The truncated propagator $\mathcal{U}_k^{(r)}$ is non-unitary and its eigenvalues lie inside the unit circle. One might think that its eigenvalues smoothly converge back to the unit circle in the unitary limit $r \to \infty$, but in many cases this is not true. If the local correlations decay exponentially one expects that the leading eigenvalue converges (or "freezes") to some value inside the unit circle, which one interprets as the leading RP resonance $\lambda_1(k)$.

For the finite-sized truncated propagator of an infinite circuit $N \to \infty$, assuming it is diagonalizable, we can illustrate that more explicitly. Let $\lambda_i^{(r)}$ be the eigenvalues of $\mathcal{U}_k^{(r)}$ (sorted by decreasing magnitude) and $\mathfrak{L}_i^{(r)} / \mathfrak{R}_i^{(r)}$ its left/right eigenvectors. A generic infinite-temperature correlation function of observables $A$, $B$ with good quasi-momentum $k$ can then be decomposed as

$$\langle A(0)B(t) \rangle = \left\langle A^\dagger, \mathcal{U}_k^t(B) \right\rangle \tag{11}$$
$$= \lim_{r \to \infty} \sum_i \left( \lambda_i^{(r)} \right)^t \left\langle A^\dagger, \mathfrak{R}_i^{(r)} \right\rangle \left\langle \mathfrak{L}_i^{(r)}, B \right\rangle$$
$$\xrightarrow[t \to \infty]{} \lambda_1^t \left\langle A^\dagger, \mathfrak{R}_1^{(\infty)} \right\rangle \left\langle \mathfrak{L}_1^{(\infty)}, B \right\rangle.$$

Therefore, provided that $A, B$ have a finite overlap with the eigenvectors associated with the isolated leading eigenvalue, their correlation function will decay as $|\lambda_1|^t$ at long times, which matches with the definition of an RP resonance in Eq. (2). Here on the LHS $\langle \bullet \rangle := \frac{1}{2^N} \operatorname{tr}(\bullet)$ (i.e., an infinite-temperature average in a finite system) and, for clarity, $A, B$ are assumed to be traceless. If $A, B$ are not traceless (i.e., they overlap with identity, which is also the eigenvalue of $\mathcal{U}$), a similar argument can be made for the connected correlation function defined in Eq. (1). In all the following sections we shall discuss connected infinite-temperature autocorrelation functions, we will usually shorten our language and simply say correlation functions. A similar procedure can also be done at finite temperature or chemical potential by introducing an appropriate inner product with a non-trivial measure.

If a circuit has a conserved quantity, correlation functions of observables overlapping with the conserved quantity will converge to a non-zero value. In the RP resonance language, this would imply the presence of $\lambda_i = 1$ in the decomposition in Eq. (11). Additionally, also all powers of the conserved quantity are conserved quantities, although typically non-local. While they cannot cause correlation functions of extensive (or local) observables to converge to a non-zero value in the thermodynamic limit, they can introduce plateaus in finite systems that decay as a power law in system size. We discuss this effect in Appendix D.

## 2.3 Example: Brickwall quantum circuits with no conserved quantities

Let us demonstrate the above RP formalism with a simple example of a circuit with no conserved quantities. The main idea is to compare the exactly calculated correlation function in a large finite system with the asymptotic decay obtained from the truncated propagator. In addition, we will show that the quasi-momentum dependence of $\lambda_1(k)$ can be non-monotonous, specifically, the slowest decay can happen at nonzero $k$.

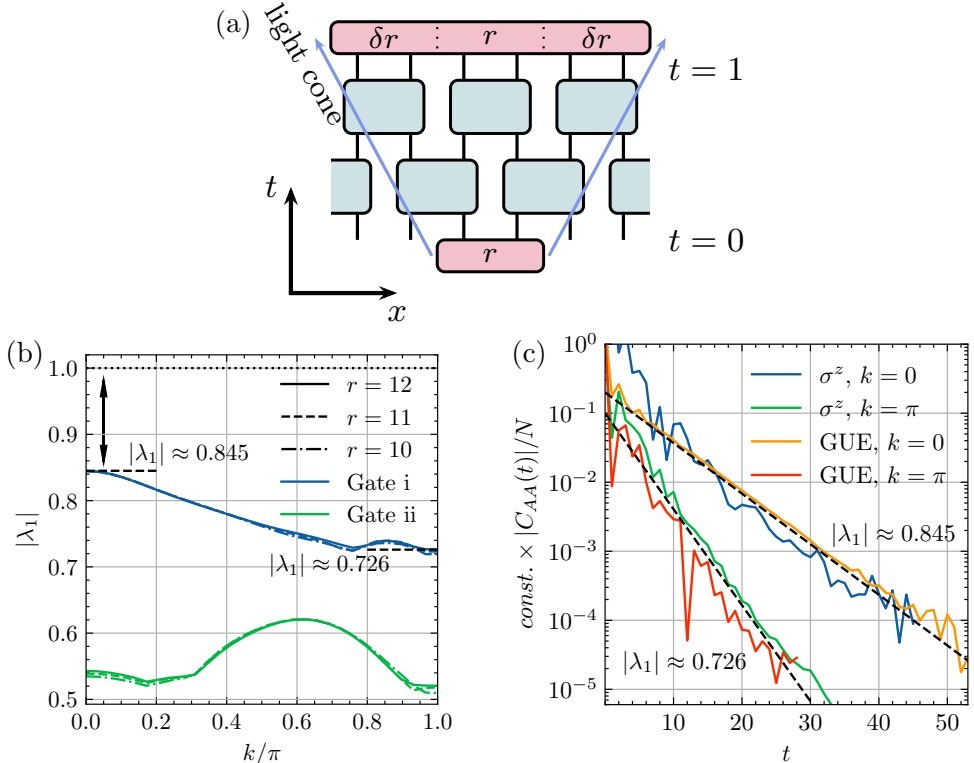

Figure 2: Circuits with no conserved quantities. (a) Diagram of operator spreading in a brickwall circuit with the same gate $V$ acting between all nearest neighbors (the shown circuit has $s = 2$, $\delta r = 2$). (b) The leading eigenvalue of the propagator truncated to the space of extensive observables with support $r$ for two realizations of a circuit with a different gate $V$ chosen randomly according to the unitary Haar measure. (c) Infinite temperature autocorrelation functions of extensive observables $A$ (4) in the circuit with gate i from (b). The local density $a$ is either $\sigma^z$, or chosen randomly according to the Gaussian unitary ensemble (GUE) [36] with support $r = 2$. Dashed lines show the RP prediction $\propto |\lambda_1|^t$, while full curves are an exact calculation in a circuit with $N = 32$ qubits; the $\sigma^z, k = 0$ case is multiplied by 15 for better presentation.

We take a brickwall circuit with nearest-neighbor 2-qubit gates, all being the same. The gate that is repeated in the circuit is picked randomly according to the unitary Haar measure. In Fig. 2b we show the leading RP resonance $|\lambda_1(k)|$ for two realizations. For details on numerical methods see Appendix B.1. The circuit's Floquet propagator can be written as

$$U = U_{\text{even}}U_{\text{odd}}, \tag{12}$$
$$U_{\text{odd}} = V_{1,2}V_{3,4} \ldots V_{N-1,N},$$
$$U_{\text{even}} = V_{2,3}V_{4,5} \ldots V_{N-2,N-1}V_{N,1},$$

where $V_{i,j}$ are local gates, i.e., 2-site unitary operators acting on sites $i$ and $j$. All $V$ are taken to be the same. In both realizations of a circuit we see that the $k$ dependence of $\lambda_1(k)$ is highly non-trivial and depends strongly on the choice of $V$. A similarly complicated behavior was also observed for the kicked Ising model in Ref. [22]. We show the $k$ dependence only for $k \in [0, \pi]$, since $|\lambda_1(k)| = |\lambda_1(-k)|$. This is a consequence of the fact that $\left(\mathcal{U}_k^{(r)}\right)^* = \mathcal{U}_{-k}^{(r)}$, which follows from the form in Eq. (10). One must only show

that $\left\langle \mathcal{S}^{sj+m}(b), \mathcal{U}\left(\mathcal{S}^{m'}(b')\right)\right\rangle$ is a real number, which quickly follows from the hermiticity of the Pauli basis, $b = b^{\dagger}, b' = b'^{\dagger}$.

Fig. 2c shows the decay of correlation functions of extensive observables with $k = 0$ and $\pi$. For details about numerical calculation of correlation functions see Appendix B.2. Their long-time decay rate matches well with the leading RP resonance determined from the truncated propagator. If the observables do not have good quasi-momentum $k$, for instance strictly local observables, their correlation function expands over the whole spectrum of eigenvalues with different momenta $k$. The long time decay is then governed by the biggest $\lambda_1(k)$. It is often assumed that this happens at $k = 0$, i.e., for translationally invariant observables, but as we see in Fig. 2b for the circuit with gate ii, this is not always the case. There the gap seems to be the smallest around $k \approx 0.6\pi$.

In the following sections we shall focus on the main subject of our work, namely, study how a conserved quantity changes the RP spectrum and what can we extract from it. As mentioned in Sec. 1.1, RP resonances can also be studied by introducing Lindbladian dissipation [15], though we find the truncated propagator more convenient; we briefly discuss the Lindblad approach in Appendix C.

## 3 The leading Ruelle-Pollicott resonance

We now focus on the main result of this paper, the effect of a single conserved quantity on the leading RP resonance. In order to study this, we need a model with only 1 conserved quantity. A natural choice for the conserved quantity in the context of spin-$\frac{1}{2}$ chains is the magnetization

$$M := \sum_j \sigma_j^z. \tag{13}$$

Therefore, our system, in addition to a translational symmetry, also has a U(1) symmetry. A natural way to achieve that would be to take a brickwall circuit with a 2-qubit gate that conserves magnetization. However, if the gate is the same everywhere, such circuits are all special – they are integrable [25, 26], including for non-brickwall geometries [37]. The second possibility that also preserves translational symmetry would be to take a brickwall configuration with two different magnetization-conserving gates, one in each brickwall layer, though it turns out that such circuits have slow (possibly non-diffusive) thermalization [38].

Because we want good diffusive thermalization, we therefore consider one of the next simplest options, which are qubit circuits with 3-site gates. The gates are arranged in a 3-layer brickwall-like geometry with $s = 3$ site translational invariance, see Fig. 3a. The Floquet propagator is given by the following expression

$$U = U_3 U_2 U_1, \tag{14}$$

$$U_i = \prod_{j=0}^{N/3} V_{i+3j,i+3j+1,i+3j+2}$$

with periodic boundary conditions (indices are taken modulo $N$). 3-site gates $V$ are taken the same everywhere, so that we have translational symmetry. In order for the circuit to conserve magnetization, the local gate $V$ must conserve it as well. One can thus quickly deduce that the local gate $V$ must have 4 independent blocks, which are shown on the diagram in Fig. 3a. To get a typical representative of such circuits, we choose the matrix elements in each block independently according to the unitary Haar measure in the corresponding block, which results in a unitary $V$.

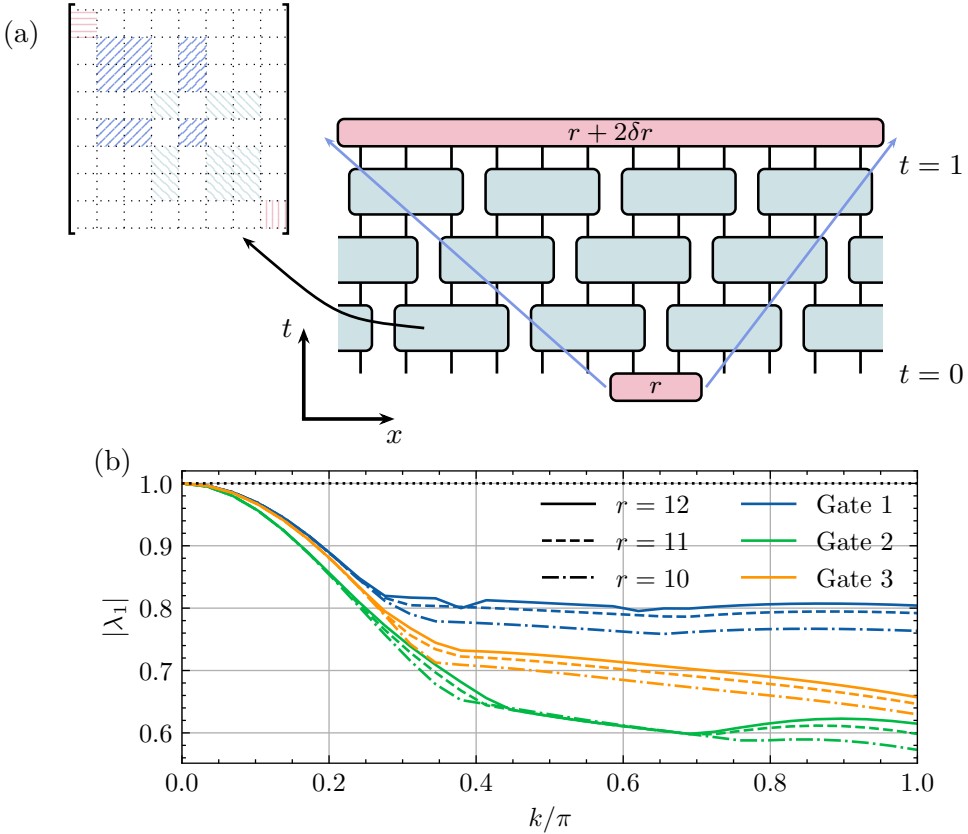

Figure 3: Circuits with conserved magnetization. (a) Circuit diagram with magnetization-conserving 3-qubit gate $V$ (the shown circuit has $s = 3$ and $\delta r = 6$), whose block structure is shown on the left. (b) The leading RP eigenvalue of the truncated propagator with support $r$ for 3 choices of $V$ (chosen according to the Haar measure). The same gates will be used throughout the paper.

The considered circuit spreads observables by $\delta r = 6$ in one period (see Fig. 3) and, importantly, we have $\delta r > s$. Therefore, Eq. (10) for the matrix elements of the truncated propagator is not applicable. While one can use the more general formula derived in Appendix A.2, this approach gives access only up to $r = 6$, since the evaluation of matrix elements is performed numerically by the evolution of observables on $r + 2\delta r$ sites. A substantial improvement can be made by noticing that successive layers are translations of the previous layer, that is $\mathcal{S}(U_i) = U_{i+1}$. This space-time symmetry [39] allows us to redefine the first layer (and a translation to the left) to be the "new" Floquet propagator for a three times shorter period, leading to less spreading $\delta r = 2$, allowing us to use Eq. (10) and thereby enabling us to reach about $r = 13$. For details see Appendix A.3.

The leading eigenvalue of the truncated propagator (and thus the leading RP resonance) for 3 different realizations with different gate $V$ is shown in Fig. 3b. We will use the same gates throughout this paper. Its $k$ dependence has two regimes, $k$ close to 0 (small $k$) and $k$ far from zero (around $k = \pi$). At $k = 0$, the leading eigenvalue is exactly 1 and its eigenvector is the local density corresponding to magnetization (i.e., the conserved quantity $a = \sigma^z$). Eigenvalues at small quasi-momenta have smooth dependence on $k$ and converge fast with $r$. We will show that they are related to transport and contain information about the spin diffusion constant in Sec. 3.1. Eigenvalues far from $k = 0$ converge slower and show more varied behavior, similarly to what we have seen in circuits without symmetries (cf. Fig. 2b), including non-smooth dependence due to collisions with

smaller eigenvalues. In Sec. 3.2 we will numerically verify that a finite gap there indeed signals exponential decay of corresponding observables that are unrelated to transport.

## 3.1 Small quasi-momenta and magnetization transport

In generic, i.e., chaotic, quantum systems with 1 conserved quantity, one typically expects diffusive transport [4]. Diffusive transport can be probed by looking at the correlation function of the local density of the conserved quantity, also called the dynamic structure factor. At sufficiently large $t$, it is expected to obey the diffusion equation [40], which gives

$$\langle \sigma^z(x,t)\sigma^z(0,0)\rangle = \frac{1}{\sqrt{4\pi Dt}}\mathrm{e}^{-\frac{x^2}{4Dt}}, \tag{15}$$

where $D$ is the model-dependent (spin) diffusion constant. We have adjusted our notation to $\sigma^z(x,t)$ meaning the $z$-Pauli matrix acting on space coordinate $x$ and propagated to time $t$. For clarity, we can also consider space to be continuous with $\Delta x = 1$ corresponding to three sites of our model (i.e., we average our system over one translational period).

Assuming Eq. (15) holds, one can derive the correlation function of magnetization with quasi-momentum $k$, $M_k := \sum_j \mathrm{e}^{-\mathrm{i}kj/3}\sigma^z_j$ [factor 3 is included to match the definition in Eq. (5)]. That is,

$$\begin{aligned}
\frac{1}{N}\left\langle M_k(t)M_k(0)^\dagger\right\rangle &= \frac{1}{N}\sum_{j,j'}\mathrm{e}^{-\mathrm{i}k(j-j')/3}\left\langle \sigma^z_j(t)\sigma^z_{j'}(0)\right\rangle \\
&= \sum_j \mathrm{e}^{-\mathrm{i}kj/3}\left\langle \sigma^z_j(t)\sigma^z_0(0)\right\rangle \\
&\to \int \langle \sigma^z(x,t)\sigma^z(0,0)\rangle\,\mathrm{e}^{-\mathrm{i}kx}\,\mathrm{d}x = \mathrm{e}^{-Dk^2 t}.
\end{aligned} \tag{16}$$

The correlation function of magnetization in momentum space therefore decays exponentially, with the decay time diverging as $1/(Dk^2)$ in the long wavelength limit $k \to 0$. This is a standard indicator of diffusion [40]. Because there are evidently exponentially decaying correlation functions with their rate having quadratic $k$ dependence, this implies that there must exist an RP resonance

$$|\lambda_1(k)| = \mathrm{e}^{-Dk^2}. \tag{17}$$

This can alternatively be interpreted as an RP continuum governing the decay of the local magnetization correlation function. We discuss other potential RP continuums in greater detail in Sec. 4.2.

Since the RP resonance predicted in Eq. (17) converges to 1 as $k \to 0$, it must be the leading RP resonance for small $k$. It is, therefore, precisely the RP resonance observed for small $k$ in Fig. 3b. This is confirmed numerically by fitting a function $\mathrm{e}^{-Dk^z}$ to $\lambda_1$ for small $k$, where $z$ is usually referred to as the transport dynamical exponent. This is shown in Fig. 4, insets (i). The obtained values of $z$ match the predicted diffusive $z = 2$ up to accessible precision. One can also check the corresponding eigenvectors, which, according to the calculation in Eq. (16), should overlap with the local density of magnetization. We have confirmed that this is indeed the case. It would be interesting to extend this to a type of non-diffusive transport ($z \neq 2$) with a one-parameter scaling relation $x^z \sim t$. We expect the quasi-momentum dependence of $\lambda_1(k)$ for small $k$ to also hold information about $z$ in that case, although the exact form of $\lambda_1(k)$ depends not just on $z$, but also on the scaling function of the local correlation function of the conserved quantity.

The fit also extracts the spin diffusion constant $D$ of the model. In order to verify that this is indeed the correct value, we compare it to the diffusion constant obtained from a

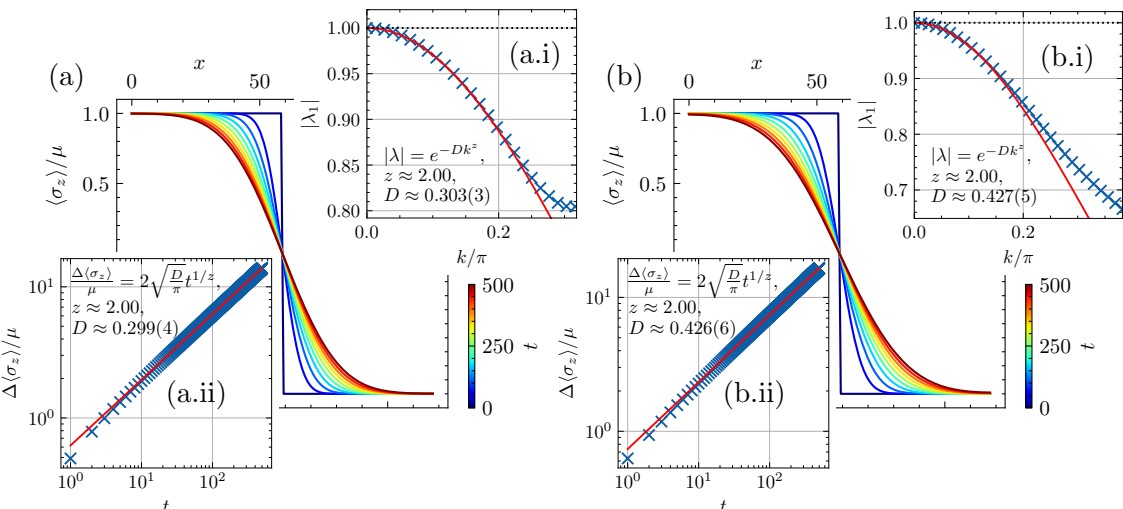

Figure 4: Transport of magnetization in the circuit with gate 1 (a) and gate 2 (b). In both instances we show the leading RP resonance ($r = 11$) and the fit to $\mathrm{e}^{-Dk^z}$ (insets i), and the domain-wall evolution (TEBD, $N = 354$ with $\mu = 10^{-3}$ and $\chi = 256$) at times with equal spacing $\Delta t = 55$ up to $t = 500$ (main plots) with the associated transferred magnetization and the diffusion constant fit (insets ii).

completely different method, namely, from the evolution of a weakly-polarized domain-wall state [41]. We prepare the system in the state with the following density matrix

$$\rho \propto \prod_{j=1}^{N/2} \mathrm{e}^{\mu\sigma_j^z} \prod_{j=N/2+1}^{N} \mathrm{e}^{-\mu\sigma_j^z}, \tag{18}$$

for small $\mu$, and evolve it with $U$. The diffusion equation dictates that the transferred magnetization $\Delta \langle \sigma^z \rangle_\rho$ should be (after long enough time)

$$\Delta \langle \sigma^z \rangle_\rho (t) := \sum_{j=N/2+1}^{N} \langle \sigma_j^z(t) - \sigma_j^z(0) \rangle_\rho = 2\mu\sqrt{\frac{D}{\pi}t}, \tag{19}$$

where $\langle \bullet \rangle_\rho := \mathrm{tr}\,(\rho \bullet)$ denotes the expectation value in the state $\rho$. The dynamics can be simulated using time-evolving block decimation (TEBD) [42, 43], for details see Appendix B.3. Fitting a curve scaling as $t^{1/z}$ to $\Delta \langle \sigma^z \rangle$ one can numerically confirm diffusion and extract the spin diffusion constant $D$.

Both methods of determining $D$ are shown in Fig. 4. In the transferred magnetization from the domain wall (bottom left (ii) insets), we again see $z$ matching the expected $z = 2$. The diffusion constants determined from different approaches also match up to accessible precision of about 1%. We have thus shown the leading RP resonance for small $k$ truly does contain information about transport. A natural next question, which we leave for future work, is how competitive is the extraction of $D$ through RP resonances compared to the current state-of-the-art.

## 3.2 Quasi-momenta far from zero and correlation function decay

The leading RP resonances far from $k = 0$ do not follow $\mathrm{e}^{-Dk^2}$ anymore and are not connected to diffusive transport, see Fig. 3b. We thus expect them to merely be indicators of exponential decay, like in the case without symmetries in Fig. 2. We demonstrate this

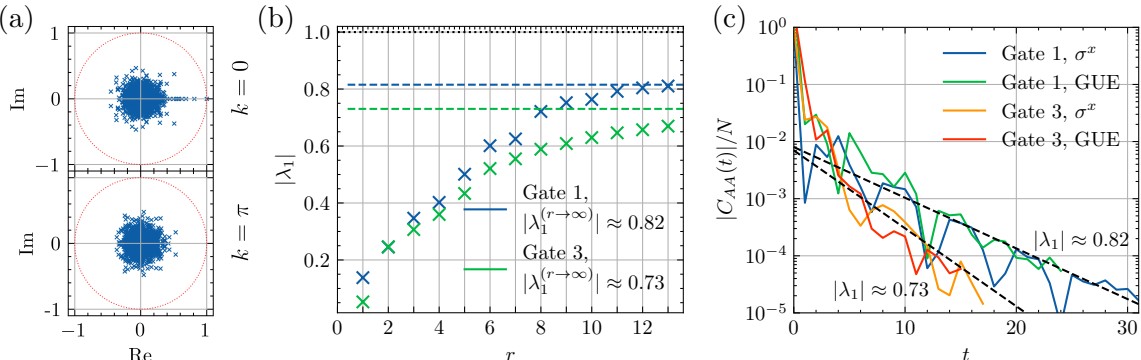

Figure 5: Leading RP resonance and correlation function decay in the $k = \pi$ spectrum for magnetization-conserving circuits. (a) The whole spectrum of the truncated propagator in $k = 0, \pi$ sectors for the circuit with gate 3 and $r = 7$. (b) Convergence of the leading RP resonances in the $k = \pi$ sector with $r$, with dashed lines denoting an extrapolation to the $r \to \infty$ limit. (c) Infinite temperature autocorrelation functions of extensive observables $A$ defined in Eq. (5) with $k = \pi$. The local densities are either $\sigma^x$ (i.e., $x$ magnetization) or chosen randomly according to the Gaussian unitary ensemble (GUE) [36] with support $r = 2$. The correlation functions are calculated in a finite circuit with $N = 30$ sites. Dashed lines show the RP predictions $\propto |\lambda_1|^t$.

numerically for $k = \pi$ in Fig. 5. Convergence with $r$ is slower than at small $k$ and at the largest available $r = 13$ the leading eigenvalue has not yet fully converged, however, it does seem that the extrapolated values[3] are strictly smaller than 1. Therefore, generic correlation functions are expected to decay exponentially with a finite rate. In Fig. 5c we show correlation functions of $\sigma^x$ magnetization and of a random 2-site observable. While exponential decay at available system size $N = 30$ is not super nice, the decay is compatible with the extrapolated $\lambda_1$, e.g., it is faster for the circuit with smaller $|\lambda_1|$. Also, different observables decay with the same rate.

The fact that the exponential decay in the shown circuits is not as evident from numerics is compatible with the slow convergence of the RP resonance with $r$. Namely, the leading RP resonance at truncation $r$ can be intuitively understood to govern the decay of correlations when the operator has spread to $r$ sites. Slower convergence of RP resonances with $r$ thus implies "nice" exponential decay only at later times and consequently smaller values of $C_{AA}$. Therefore, it is expected that exponential decay will be harder to see numerically at finite $N$ precisely when RP resonances will also be hard to extract numerically. Additionally, correlation functions in Fig. 5c visibly oscillate, making numerical checks less clear. This, however, is merely a consequence of $\lambda_1$ having an imaginary component seen for the example of gate 3 in Fig. 5a.

## 4   Beyond the leading Ruelle-Pollicott resonance

We have shown that the leading RP resonance $\lambda_1(k)$ for small $k$ is connected to transport of the conserved quantity. How about subleading eigenvalues; they could contain information about other correlation functions. One might expect that correlation func-

---

[3]The extrapolation is done through an exponential fit. We have no good theoretical justification for the exponential fit, it was simply the best match for the data.

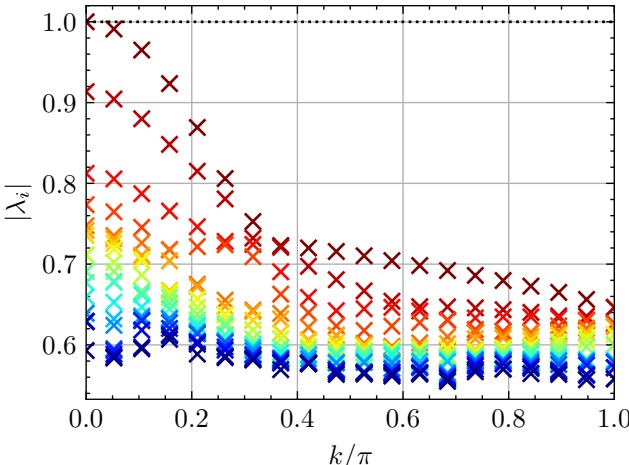

Figure 6: 20 leading eigenvalues of the truncated propagator at different $k$. Calculated for the circuit with gate 3 and support $r = 11$. The subleading eigenvalues are not yet converged at accessible $r$, see the discussion in the main text and Fig. 7 for details.

tions of observables orthogonal to all transport-related quantities decay exponentially, like in a system without any conserved quantities, which would imply that there must exist subleading RP resonances.

The first few largest eigenvalues of the truncated propagator can be calculated numerically and are shown in Fig. 6. We observe many subleading eigenvalues in a narrow interval and one might think that they are isolated RP resonances. It turns out this is not the case; these eigenvalues have not yet converged with $r$ and, moreover, many of them look to be converging to the leading eigenvalue $\lambda_1$ (see Fig. 7 for the convergence to $|\lambda_1| = 1$ in the $k = 0$ sector). We shall propose in Sec. 4.1 that in the $k = 0$ sector, some of such eigenvalues close to 1 come from the powers of the conserved magnetization $M$. Furthermore, we shall argue in Sec. 4.2 that power-law hydrodynamic tails in transport-related correlation functions imply an RP continuum in the spectrum of the truncated propagator as sketched in Fig. 1b. Both of these results make it hard to numerically determine whether subleading discrete RP resonances exist.

## 4.1 Powers of the conserved quantity

Eigenvectors corresponding to $\lambda = 1$ are conserved quantities and while our system does have only 1 local conserved quantity – the magnetization $M$ – all its powers are also conserved. Because our truncated operator propagator is geared towards local operators, while $M^2 = \sum_{i,k} \sigma_i^z \sigma_k^z$ is non-local, it does not have an eigenvalue $\lambda = 1$ that would correspond to $M^2$. However, as we will now argue[4], it does have an eigenvalue that is close to 1.

Without truncation $M^2$ is conserved. This can be also seen explicitly, $\mathcal{U}(M^2) = \sum_{i,k} \mathcal{U}(\sigma_i^z)\mathcal{U}(\sigma_k^z)$, which, after invoking definition of the spin current $j_k$ [see Eq. (21)], $\mathcal{U}(\sigma_k^z) = \sigma_k^z + j_{k-1} - j_k$, can be rewritten as $\mathcal{U}(M^2) = \sum_{i,k}(\sigma_i^z + j_{i-1} - j_i)(\sigma_k^z + j_{k-1} - j_k) = M^2$. Conservation comes about because the many-qubit terms $j_i j_k$ (and likewise $\sigma_i^z j_k$) sum to 0 due to alternating signs stemming from the continuity equation.

In the basis truncated to support $r$, some of the terms needed for cancellation are

---

[4]We thank Rustem Sharipov for this suggestion.

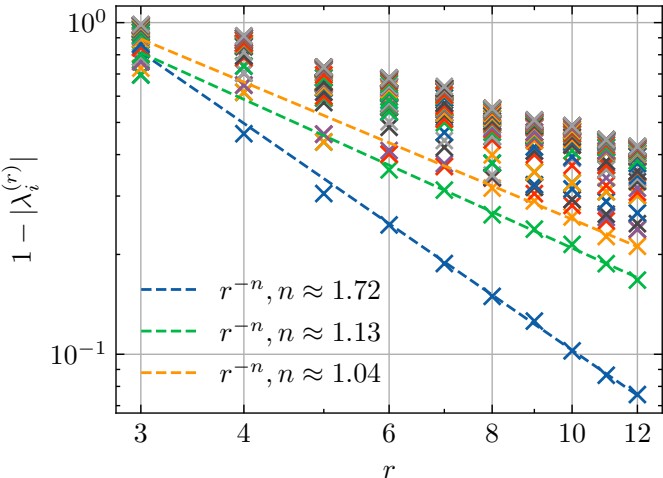

Figure 7: Convergence of 50 subleading eigenvalues of the truncated propagator in the $k = 0$ sector to the unit circle. A power-law fit for the 3 subleading eigenvalues is shown.

missing, resulting in an almost-conservation. Let us denote the truncated $M^2$ by $Q^{(r)}$, explicitly $Q^{(r)} = \sum_{i,k,|k-i|<r} \sigma_i^z \sigma_k^z$. In particular, the $j_i j_k$ terms coming from $\mathcal{U}(\sigma_i^z \sigma_k^z)$ with $|k - i| = r - 1$, that would cancel out with the same (but negative) terms from $\mathcal{U}(\sigma_i^z \sigma_{k+1}^z)$, do not cancel out, since $\sigma_i^z \sigma_{k+1}^z$ is not included in $Q^{(r)}$ (and similarly for $\sigma_i^z j_k$ terms). Importantly, one can check that for $|k - i| < r - 1$ the cancellation still occurs and all such terms are conserved. The upshot is that the only terms differing between $Q^{(r)}$ and $\mathcal{U}(Q^{(r)})$ are those coming from $\sigma_i^z \sigma_k^z$ with $|k - i| = r - 1$, the number of which does not scale with $r$.

The number of terms in $Q^{(r)}$ is equal to the number of indices with $|k - i| < r$, i.e., it scales as $\sim r$ ($\sim Nr$ in a system of length $N$). On the other hand, the number of term differing between $\mathcal{U}(Q^{(r)})$ and $Q^{(r)}$ is constant, $\sim c$. In the limit $r \to \infty$ the relative difference in small. Therefore, this suggests that $Q^{(r)}$ is almost an eigenvector with eigenvalue being close to 1, which we estimate to be[5]

$$|\lambda| \approx 1 - \frac{\left\| \mathcal{U}(Q^{(r)}) - Q^{(r)} \right\|}{\left\| Q^{(r)} \right\|} = 1 - \frac{c}{r}. \tag{20}$$

We expect this eigenvalue also in the spectrum of the truncated propagator $\mathcal{U}_{k=0}^{(r)}$. A similar informal calculation can be done for an arbitrary power of magnetization.

Numerical tests of Eq. (20) by calculating 50 subleading eigenvalues of the truncated propagator are shown in Fig. 7. A few largest subleading eigenvalues indeed converge to the unit circle according to a power law. The first subleading eigenvalue $\lambda_2$ in fact seems

---

[5] More exactly, given a diagonalizable matrix $M$ and any vector $v$, we have $\|Mv\| = \left\| P^{-1} \Lambda P v \right\| \leq |\lambda_v| \|P\|_{\mathrm{op}} \left\| P^{-1} \right\|_{\mathrm{op}} \|v\|$. Here, $\Lambda$ is the diagonal matrix of eigenvalues, $P$ the transition matrix into the eigenbasis of $M$, $\|\bullet\|_{\mathrm{op}}$ the operator norm corresponding to $\|\bullet\|$ and $\lambda_v$ the largest eigenvalue (by magnitude) such that the corresponding eigenvector overlaps with $v$. The inequality gives $|\lambda_v| \geq \frac{1}{\|P\|_{\mathrm{op}} \|P^{-1}\|_{\mathrm{op}}} \frac{\|Mv\|}{\|v\|}$, which differs from Eq. (20) only by the factor of $\frac{1}{\kappa(M)} = \frac{1}{\|P\|_{\mathrm{op}} \|P^{-1}\|_{\mathrm{op}}}$. $\kappa(M)$ is the spectral condition number and measures the non-normality of a matrix (for normal matrices $\kappa = 1$, for non-normal $\kappa > 1$). For the truncated propagator one can numerically check that $\kappa$ is large and increases with $r$, making the bound uninformative. Nevertheless, the numerical evidence in Fig. 7 suggests that the informal estimate (20) still roughly holds. This can be traced back to the empirical fact that the subleading eigenvector is well behaved and almost equal to $Q^{(r)}$.

to converge even faster than the predicted $1/r$. This might be a consequence of small $r$ accessible numerically. Further analysis reveals that the eigenvector corresponding to $\lambda_2$ has a large overlap with $Q^{(r)}$. Similarly, the second subleading eigenvector (corresponding to $\lambda_3$) seems to have substantial overlaps with the truncation of $M^3$.

The truncated propagator in $k = 0$ therefore seems to have a number of eigenvalues that converge to 1 as $1/r$ that come from powers of $M$. With the truncation that we use it is, therefore, hard to numerically see if there are any genuine isolated subleading resonances. Furthermore, it is not clear if subleading RP resonances should even exist due to power-law hydrodynamic tails in correlation functions, as we discuss in the next section.

## 4.2 Hydrodynamic tails

In systems with conserved quantities certain correlation functions related to transport exhibit power-law decays, i.e., the so-called hydrodynamic tails [40, 44], due to slow power-law spreading of inhomogeneities of the conserved density (distance and time scaling as $x^z \sim t$). A typical example is the correlation function of local magnetization (15), i.e., the Green's function of the diffusion equation, decaying as $\sim 1/t^{1/2}$ at $x = 0$. Similar power-law tails occur also in higher-order corrections to diffusion equation [40, 44], more recently studied in the context of effective field theories [45–48].

Power-law decay $C(t) \sim t^{-\alpha}$, for some $\alpha > 0$, being slower than exponential, cannot be governed by a discrete RP resonance. If we nonetheless assume it expands over a spectrum of eigenvalues as $C(t) = \sum_j c_j \lambda_j^t$, where $c_j$ are expansions coefficients [cf. Eq. (11)], it must expand over a continuum of eigenvalues (an "RP continuum") with gap closing to $\lambda = 1$[6]. This is precisely what we observed for the correlation function of magnetization in momentum space (16), with the gap closing at $k = 0$. Additionally, the quasi-momentum dependence of the RP continuum was exactly determined in that case, $|\lambda(k)| = e^{-Dk^2}$. The expansion over eigenvalues for all $k$ is expected only for correlation functions of local observables. For extensive observables, the whole continuum must lie in one $k$ sector. A potential RP continuum governing a power-law decay of a $k = 0$ extensive observable is depicted in the diagram in Fig. 8.

Power-law decay is not the only type of non-exponential decay expected in systems with one conserved quantity. One other possibility is a power-law correction to exponential decay $C(t) \sim t^{-\alpha} e^{-\nu_0 t}$, for some $\alpha$ and $\nu_0 > 0$, which is governed by a continuum $\lambda \in [0, e^{-\nu_0}]$[7], also depicted in Fig. 8. Such decays were observed in corrections to the correlation function of magnetization with quasi-momentum $\langle M_k(t) M_k(0) \rangle$ (see Eq. (16) for the exponential diffusive prediction), recently explicitly calculated with the methods of effective field theory [48]. The final possibility is a stretched-exponential decay $C(t) \sim e^{-t^\alpha}$ for some $0 < \alpha < 1$, which was recently predicted to occur for all local correlation functions unrelated to transport in 1d [49]. It is governed by a continuum $\lambda \in [0, 1]$ that we discuss further in Appendix E.

Based on the above arguments, we conjecture that there is an RP continuum of eigenvalues for all $|\lambda|$ below the leading RP resonance governing the transport of magnetization. In other words, all $|\lambda| \leq e^{-Dk^2}$ are part of an RP continuum.

---

[6]The argument can be illustrated more explicitly by writing the decomposition as an integral $C(t) = \sum_j c_j e^{-\nu_j t} = \int_0^\infty c(\nu) e^{-\nu t} \, d\nu$, where $\lambda = e^{-\nu}$, $c(\nu)$ is the continuous version of the expansion coefficients and we assumed $\lambda$ to be real for simplicity. In other words, $C(t)$ is the Laplace transform of $c(\nu)$. By employing the inverse Laplace transform, we see that $C(t) = t^{-\alpha}$ is obtained for $c(\nu) = \frac{1}{\Gamma(\alpha)} \nu^{\alpha-1} \theta(\nu)$, where $\Gamma$ is the Gamma function and $\theta$ is the Heaviside step function. We observe that $c(\nu) \neq 0$ for all $\nu > 0$ – an RP continuum over all $\lambda \in [0, 1]$.

[7]$C(t) = t^{-\alpha} e^{-\nu_0 t}$ leads to $c(\nu) = \frac{1}{\Gamma(\alpha)} (\nu - \nu_0)^{\alpha-1} \theta(\nu - \nu_0)$ via the inverse Laplace transform.

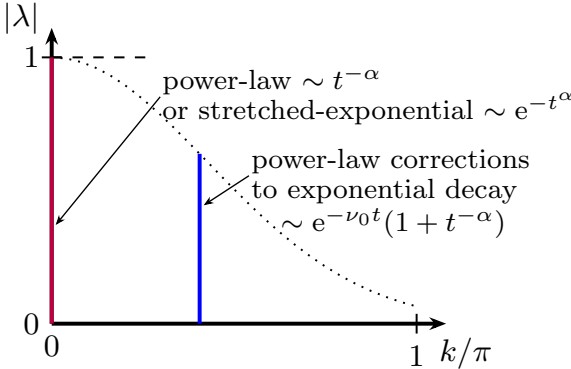

Figure 8: Diagrams of possible RP continuums and the type of correlation function decay of extensive observables they govern.

### 4.2.1  Extensive spin current

Power-law tails are expected also in the spin current $j$ defined via the (discrete space and time) continuity equation

$$\mathcal{U}(M_{[i,l]}) - M_{[i,l]} = j_{i-1} - j_l, \tag{21}$$

where $j_i$ is the current operator between sites $i$ and $i+1$, and we introduced the magnetization acting on sites $i$ to $l$ as $M_{[i,l]} := \sum_{j=i}^{l} \sigma_j^z$. The closed-form expression for $j_i$ is complicated to derive analytically for our 3-layer brickwall circuit, but can be obtained numerically for a given gate by evaluating the LHS of Eq. (21) in a big enough finite circuit. Current densities $j_i$ are different for different values of $i$ modulo 3, and have support on at most 9 sites. For a 2-layer brickwall circuit (e.g., Fig. 2a) the expressions are more manageable (2 or 4 site operators) and are written out for any magnetization-conserving gate in Ref. [26].

The asymptotics at large $t$ of the local spin current correlation function can be deduced from the local magnetization correlation function and the continuity equation. The diffusive local magnetization correlation function (15) (i.e., the diffusion equation Green's function) implies

$$\langle j(x,t)j(0,0)\rangle = \frac{x^2 - 2Dt}{8\sqrt{\pi D}t^{5/2}} e^{-\frac{x^2}{4Dt}}. \tag{22}$$

For the derivation see Appendix F. At $x = 0$ this again decays as a power-law $\sim 1/t^{3/2}$.

Of particular interest is the extensive current

$$J := \sum_i j_i, \tag{23}$$

because it appears in the Green-Kubo equation for the diffusion constant [50][8]. Using Eq. (22) and the fact that the equilibrium correlation function is translationally invariant in space, we get for the leading order of the extensive current in the thermodynamic limit

$$\lim_{N\to\infty} \frac{1}{N} \langle J(t)J(0)\rangle = \int_{-\infty}^{\infty} \langle j(x,t)j(0,0)\rangle \, \mathrm{d}x = 0. \tag{24}$$

---

[8]For discrete-time dynamics, the Green-Kubo formula reads $D = \lim_{t_{\max}\to\infty} \lim_{N\to\infty} \frac{1}{N}\left(\frac{1}{2}\langle J(0)J(0)\rangle + \sum_{t=1}^{t_{\max}} \langle J(t)J(0)\rangle\right)$. In the considered circuits, it gives $D \approx 0.31$ in the circuit with gate 1 and $D \approx 0.44$ in the circuit with gate 2, which matches our predictions with other methods up to accessible precision (see Fig. 4).

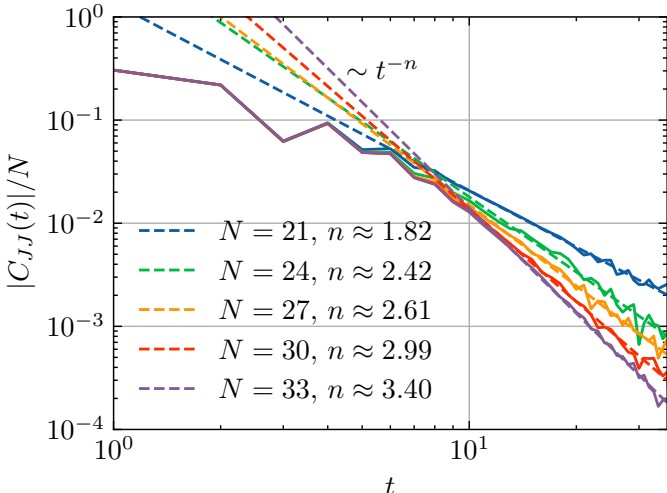

Figure 9: Infinite temperature autocorrelation of extensive spin current. The correlation functions are calculated in circuits with gate 3 with a different number of sites $N$. Dashed lines show power-law fits.

The long-time prediction coming from the diffusive (i.e., first order) local magnetization correlation function, thus gives exactly 0, which can be traced back to the fact that $\langle j(x,t)j(0,0)\rangle$ is a total derivative in $x$ [see Eq. (22)][9]. This is very interesting as it means that the leading asymptotic behavior of the correlation function of the extensive current is not $\sim t^{-3/2}$ (in 1d), but is rather given by subleading corrections to the local magnetization correlation function. Correlation function of $J$ is, therefore, a sensitive probe of subleading corrections to diffusion, that would perhaps be hard to see in other correlation functions that are nonzero already in the leading order.

Numerical calculation of the extensive current correlation function for one circuit realization is shown in Fig. 9. It decays as a power law for any finite system size $N$, however, the power increases with $N$. It is not clear what will happen in the thermodynamic limit; its power-law dependence may converge to some finite power or the decay may become exponential.

In addition to the inconclusive numerics, it is also not clear what is theoretically expected. Power-law decay $\sim t^{-3/2}$ is expected in systems with an additional conserved quantity, e.g., in Hamiltonian systems [44], and can be explained by the cross transport of the conserved quantities. In systems with only one U(1) conserved quantity a power-law decay with a higher power was predicted [48], although exponential decay is also sometimes observed, e.g., in the kicked Ising model [51]. The power law predictions are based on the corrections to the diffusive local magnetization correlation function calculated from the effective field theory of diffusion [47], however, from the calculation in Ref. [48] it is not clear what is the first nonzero correction contributing to the extensive current correlation function. Furthermore, subleading corrections can depend on unknown model-dependent parameters.

All in all, the fundamental question of the asymptotics of the extensive current correlation function $C_{JJ}$ still remains. Is it a power law, like the local current correlation function, or are the hydrodynamic tails absent and, if they are, when and why?

---

[9]Note that, interestingly, according to the Green-Kubo formula, the diffusion constant $D$ must acquire its nonzero value at times before the local magnetization correlation function (15) gets its Gaussian form and $\langle J(t)J(0)\rangle$ becomes zero.

# 5 Conclusion

We have studied the (leading) Ruelle-Pollicott resonances in translationally invariant qubit circuits with 3-site magnetization-conserving gates. The resonances were obtained from the (leading) eigenvalues of the truncated operator propagator in the limit of no truncation. Our main result is that the quasi-momentum dependence of the leading Ruelle-Pollicott resonance $\lambda_1$ can be used to extract the transport properties. Specifically, for diffusive systems, we have $|\lambda_1(k)| = e^{-Dk^2}$ for small quasi-momenta $k$, allowing us to numerically determine the spin diffusion constant $D$. For large $k$, the leading resonance is not related to transport and governs the decay of respective correlation functions. We also argue for the existence of a continuum of eigenvalues below the leading diffusive resonance, $|\lambda| < e^{-Dk^2}$, which controls non-exponentially decaying correlation functions, for instance, due to power-law hydrodynamic tails.

Even though our analysis was limited to a specific qubit circuit model, we expect our conclusions about the Ruelle-Pollicott resonances to be generic for systems with exactly one U(1) conserved quantity. In particular, the final expression for the matrix elements of the truncated propagator in Eq. (10) [or, in the case of larger operator spreading, Eq. (28)] hold for any quantum circuit, for example, with a different arrangement of gates or higher local dimension. An interesting open question is generalizing the truncated propagator to Hamiltonian systems.

Another compelling direction is benchmarking the truncated propagator method for extracting the diffusion constant against other standard methods. While we have shown that the Ruelle-Pollicott method produces the same results as a tensor-network simulation and the Green-Kubo formula, it is not clear whether and when one method might be preferable to others[10]. Additionally, the leading Ruelle-Pollicott resonance can hold information about non-diffusive transport, i.e., ballistic, sub- or super-diffusive. However, the exact dependence of the leading resonance on $k$ depends on the asymptotic (space and time) behavior of the correlation function of the local density of the conserved quantity (i.e., the scaling function) and might be complicated. This can hinder the effectiveness of identifying the transport type and extracting model-dependent constants. Especially interesting are integrable systems with an extensive number of conserved quantities, where we expect an extensive number of RP continuums governing transport.

A focus of future work can also be to better understand the conjecture about subleading Ruelle-Pollicott continuums. A promising direction seems to be developing a different truncation scheme that separates asymptotically differently behaving observables, i.e., those which are governed by hydrodynamics and those which are not. This can also shed additional light on the stretched-exponential decay recently predicted for correlation functions of local non-hydrodynamic observables in 1d systems [49]. A particularly interesting physical question in this context is also the asymptotic behavior of the extensive current correlation function. We have shown that it is governed by the corrections to the diffusive correlation function of local magnetization recently argued to be universal for systems with exactly one U(1) conserved quantity [48].

---

[10]While a detailed analysis would be nice, it seems that the RP approach has some advantages compared to the Lindblad-based approaches with dissipation targeting long Pauli strings, like the DAOE [35]. Namely, we work directly in the infinite-size limit and do the truncation instantly and explicitly. For instance, for a particular system in Fig. 7 of Ref. [34] the RP approach gives the diffusion constant that is within less than 1% of what is believed to be the correct value, while the DAOE is about 3% off (and the Krylov method about 5% off).

# Acknowledgements

U.D. would like to thank Rustem Sharipov, Matija Koterle, Lenart Zadnik, and Tomaž Prosen for insightful discussions. We also acknowledge support by Grants No. J1-4385 and No. P1-0402 from the Slovenian Research Agency (ARIS).

# A   Details about the quasi-momentum-dependent propagator

In this appendix, we more carefully define the mathematical structure of the space of extensive observables briefly explained in Sec. 2.1 and use it to derive the matrix elements of the truncated propagator, Eq. (10).

## A.1   Space of extensive observables

As explained in words in Sec. 2.1, the space of extensive observables of support $r$ is spanned by

$$\mathbb{P}_k^{(r)} := \bigcup_{m=0}^{s-1} \left\{ B_k^{(m,b)} \mid b \in \mathscr{P}^{(r)} \right\}, \tag{25}$$

where $B_k^{(m,b)}$ is defined in Eq. (8) and $\mathscr{P}^{(r)}$ in Eq. (6). $\mathbb{P}_k^{(r)}$ is an orthonormal basis w.r.t. the extensive Hilbert-Schmidt inner product, $\langle\!\langle A, B \rangle\!\rangle = \frac{s}{N2^N} \operatorname{tr} A^\dagger B$, defined in Sec. 2.1. This can be deduced from a direct calculation,

$$\left\langle\!\left\langle B_k^{(m,b)}, B_k^{(m',b')} \right\rangle\!\right\rangle = \tag{26}$$

$$= \frac{s}{N2^N} \operatorname{tr} \left[ \sum_j \mathrm{e}^{\mathrm{i}kj} \mathcal{S}^{sj+m}(b) \sum_{j'} \mathrm{e}^{-\mathrm{i}kj'} \mathcal{S}^{sj'+m'}(b') \right] =$$

$$= \frac{s}{N2^N} \sum_{j,j'} \mathrm{e}^{\mathrm{i}k(j-j')} \operatorname{tr} \left[ \mathcal{S}^{sj+m}(b) \mathcal{S}^{sj'+m'}(b') \right] =$$

$$= \frac{s}{N} \sum_{j,j'} \mathrm{e}^{\mathrm{i}k(j-j')} \delta_{m,m'} \delta_{j,j'} \delta_{b,b'} = \delta_{m,m'} \delta_{b,b'},$$

where $\delta$ is the Kronecker delta. We used that

$$\left\langle \mathcal{S}^j(b), \mathcal{S}^{j'}(b') \right\rangle = \delta_{j,j'} \left\langle b, b' \right\rangle \tag{27}$$

for $b, b' \in \operatorname{Span}(\mathscr{P}^{(r)})$. Here the local Hilbert-Schmidt inner product is used, $\langle a, b \rangle = \frac{1}{2^N} \operatorname{tr} a^\dagger b$, defined already in Sec. 2.1. This is a consequence of the fact that $b, b'$ act with a traceless operator on the first site. For any $j \neq j'$ we thus get a traceless operator acting on either site $j$ or $j'$, rendering the whole inner product zero. If additionally $b, b' \in \mathscr{P}^{(r)}$, we have $\langle b, b' \rangle = \delta_{b,b'}$.

## A.2   Propagator of extensive observables

We now express the Heisenberg propagator $\mathcal{U}$ in the basis $\mathbb{P}_k := \lim_{r \to \infty} \mathbb{P}_k^{(r)}$. Since we are working with an orthonormal basis, the matrix elements $[\mathcal{U}_k]_{(m,b),(m',b')}$ are simply

$$[\mathcal{U}_k]_{(m,b),(m',b')} = \left\langle\!\left\langle B_k^{(m,b)}, \mathcal{U}\left( B_k^{(m',b')} \right) \right\rangle\!\right\rangle =$$

$$= \frac{s}{N} \sum_{j,j'} \mathrm{e}^{\mathrm{i}k(j-j')} \left\langle \mathcal{S}^{sj+m}(b), \mathcal{U}\left( \mathcal{S}^{sj'+m'}(b') \right) \right\rangle$$

$$= \frac{s}{N} \sum_{j,j'} \mathrm{e}^{\mathrm{i}k(j-j')} \left\langle \mathcal{S}^{s(j-j')+m}(b), \mathcal{U}\left( \mathcal{S}^{m'}(b') \right) \right\rangle$$

$$= \sum_j \mathrm{e}^{\mathrm{i}kj} \left\langle \mathcal{S}^{sj+m}(b), \mathcal{U}\left( \mathcal{S}^{m'}(b') \right) \right\rangle, \tag{28}$$

where we took into account translational invariance and the cyclic property of trace.

In many cases, the sum in Eq. (28) has only a few non-zero terms. The reason for that is similar to the one employed in showing Eq. (27): since $b, b'$ act with a traceless Pauli operator on the first site, their translations must overlap on it in order to give a non-zero inner product. In addition to that, Eq. (28) contains propagation $\mathcal{U}\left(\mathcal{S}^{m'}(b')\right)$, which can change the first site on which $b'$ acts non-trivially.

As mentioned already at the beginning of Sec. 2, in quantum circuits, the support of observables can increase by at most a finite amount in one period, i.e., the light cone is exact. In general, the spreading of operators depends on the geometry, they can even spread to all sites in one time period, e.g., in staircase circuits. Additionally, it depends on the site at which they act, e.g., a two site operator in a brickwall circuit shown in Fig. 2 spreads by 1 site to the left and right if it begins at an odd site and by 2 sites to the left and right if it begins at an even site (the case depicted in the figure). Furthermore, it can also be different to the left and to the right, e.g., in the circuit considered in the bulk of this paper, the operator shown in Fig. 3 spreads by 6 sites to the left and 4 sites to the right. To simplify the discussion, we define $\delta r$ to be the maximum spreading for any site and any direction (left/right) in a considered circuit. One can easily see that if $\delta r \leq s$, only three terms in Eq. (28) can be non-zero, and we can simplify it to

$$[\mathcal{U}_k]_{(m,b),(m',b')} = \sum_{j=-1}^{1} e^{ikj} \left\langle \mathcal{S}^{sj+m}(b), \mathcal{U}\mathcal{S}^{m'}(b') \right\rangle. \tag{29}$$

This is precisely Eq. (10) we set out to derive. An analogous form was written down in particular cases in Refs. [21, 22, 25]. The sum has an interpretation of "realigning" propagated densities back into the basis, where the mapping between densities and extensive observables is one-to-one. For details about this interpretation see Appendix A in Ref. [24].

Eq. (29) applies to many circuits studied in the literature, such as the brickwall circuits, where $\delta r = s = 2$, see Fig. 2. It also applies to the whole set of canonical geometries with 2-site nearest-neighbor gates introduced in Ref. [39]. In the 3-site gate circuits considered in the bulk of this paper (see Fig. 3), the spreading is faster, though; there $\delta r = 6$ and $s = 3$ and therefore more than 3 terms in Eq. (28) can be non-zero. One can, however, use a trick described in the following section.

### A.3 Space-time symmetries

While matrix elements of the truncated propagator for arbitrary spreading $\delta r$ can be evaluated by Eq. (28), the expressions can sometimes be simplified by exploiting a space-time symmetry. Namely, in both the brickwall circuit defined in Eq. (12) and the 3-site generalization of the brickwall defined in Eq. (14), one can notice that successive layers (i.e., $U_{\text{odd/even}}$ or $U_{1/2/3}$) are translations of the previous layer. This can be interpreted as a space-time symmetry; translations in space are equivalent to translations by a fraction of a period in time. In equation, valid for both of the mentioned cases,

$$S^\dagger U(0,1)S = U(1/s, 1 + 1/s), \tag{30}$$

where $S$ is the 1-site translation operator to the left, $S^\dagger(a \otimes \mathbb{1})S = \mathbb{1} \otimes a$, $s$ is the number of sites of translation invariance (and, crucially, also the number of layers), and $U(t_1, t_2)$ denotes the propagator from time $t_1$ to $t_2$. The time is defined to be propagated by $1/s$ after the application of each layer, in particular the Floquet propagator is $U \equiv U(0, 1)$.

Space-time symmetries and their implications are discussed in greater detail in Ref. [39]. An important result for a large class of circuits is that their propagator can be written in

the following way

$$U = S^{-s}(S\tilde{U})^s, \tag{31}$$

where $\tilde{U}$ is a Floquet propagator of a circuit of some other (preferably simpler) geometry. In both of the considered examples $\tilde{U}$ is the propagator of the 1st layer, $\tilde{U} = U_{\mathrm{odd}}$ for the brickwall (12) and $\tilde{U} = U_1$ for its 3-site generalization (14).

Since $S^{-s}$ is just a phase in the quasi-momentum eigenbasis, one can redefine $S\tilde{U}$ to be the new equivalent Floquet propagator (i.e., the propagator for "one period"). In particular, for the 3-site generalization of the brickwall, this reduces the spreading in one period from $\delta r = 6$ to $\delta r = 2 < s = 3$, allowing us to use the simpler expression for the matrix elements of the truncated propagator, Eq. (10), and allowing for bigger numerically reachable supports (it requires working with operators supported on at most $r+2\delta r = r+4$ sites, instead of $r + 12$), see Appendix B.1 for details about the numerics. Since the long-time behavior is not affected by the redefinition of the Floquet propagator, the extracted RP resonances must be the same. One must merely account for the fact that the new Floquet propagator $S\tilde{U}$ propagates only for time $1/s$ and thus take the $s$-th power of the extracted resonances. Additionally, if one would study the phase of RP resonances, they would have to be rotated in each $k$ sector by the appropriate phase stemming from $S^{-s}$ in Eq. (31).

# B    Numerical methods

## B.1    Spectrum of the truncated propagator

RP resonances are extracted numerically by diagonalizing $\mathcal{U}_k^{(r)}$, which can be represented as a finite matrix with matrix elements explicitly given by Eq. (10). Support $r$ basis has $\frac{3s}{4}4^r$ elements [see Eq. (25)], thus resulting in a $\frac{3s}{4}4^r \times \frac{3s}{4}4^r$ matrix. Importantly, the evaluation of matrix elements is performed by propagating observables with support $r + 2\delta r$ and are, therefore, numerically represented as $\frac{3s}{4}4^{(r+2\delta r)}$ component vectors.

Exact diagonalization can be used for smaller supports, up to around $r = 7$ for our case of $s = 3$. At larger supports, we are constrained by memory and are forced to use iterative methods, which target only a finite number of eigenvalues. Since we are typically interested only in the first few leading eigenvalues, this can be done efficiently. We must only evaluate Eq. (10) on the fly, i.e., by directly acting on a given observable. The calculations are again done on observables with at most support $r + 2\delta r$, which means that the maximum reachable support is constrained also by the amount of spreading. In the considered cases, this approach gives us access to support around $r = 13$. We use Arnoldi iteration implemented in ARPACK [52].

The truncated propagator is generically a non-normal matrix (i.e., it does not commute with its Hermitian adjoint) and its eigenvectors are, therefore, not orthogonal. This can lead to numerical difficulties and subtleties when interpreting its spectrum [53]. It was already observed in Ref. [19] that RP resonance eigenvectors are singular objects, however, to what extent non-normality causes other problems when extracting RP resonances remains to be seen.

## B.2    Correlation functions

Throughout the paper we numerically calculate connected infinite temperature autocorrelation functions, that is

$$C_{AA}(t) = \langle A(t)A(0) \rangle = \frac{1}{2^N} \operatorname{tr} A(t)A(0) \tag{32}$$
$$= \frac{1}{2^N} \sum_\psi \langle \psi | A(t)A(0) | \psi \rangle,$$

where the sum is taken over some normalized basis of the Hilbert space, and $N$ is the number of sites in the circuit. For clarity, we assumed traceless $A$, although the following discussion straightforwardly generalizes to any $A$. Eq. (32) can be numerically evaluated by definition up to $N = 18$. In bigger systems, we are constrained by memory and must use the typicality approach. Namely, fluctuations of $C_{AA}(t)$ in a single random state are of the order of $\sim 1/2^{N/2}$. Therefore, the trace in Eq. (32) can be approximated by only 1 random state, resulting in error on the order of $10^{-5}$ at $N = 33$. Crucially, both $A$ and $U$ must be sufficiently local for this approach and their action on a state must be implemented efficiently. That is, the full $2^N \times 2^N$ matrices must never be saved in memory, only their action on a state must be evaluated on the fly [e.g., by Eqs (5), (12) and (14)].

## B.3    Domain wall quench

To determine the diffusion constant independently of RP resonances, we simulate unitary evolution starting with a weakly polarized domain wall state (see Sec. 3.1). In order to probe the behavior in the thermodynamic limit, it is crucial that the numerical simulation only goes up to times, where the boundary has not yet meaningfully affected the dynamics. In diffusive systems, this time can be approximated by the expected spreading of the diffusive front $t \sim x_{\max}^2/D$. To reach big enough times, we use TEBD [42,43]. The diffusion constant is expressed from the transferred magnetization in state defined in Eq. (18). According to Eq. (19), this involves evaluating

$$\left\langle \sigma_j^z(t) \right\rangle_\rho = \operatorname{tr} \rho(t)\sigma_j^z. \tag{33}$$

There are multiple ways to numerically calculate this, for example one can write either $\rho$ or $\sigma_z^j$ as a matrix product operator (MPO) and evolve them, or appropriately sample and evolve pure states. For small $\mu$, evolving $\rho$ turns out to have the slowest growth of Schmidt coefficients and is therefore the most efficient to simulate. This is compatible with previous findings in similar systems, for details see Ref. [41]. We used this method in the main part of the paper setting the maximum bond dimension to $\chi = 256$. Additionally, we checked the results indeed converged by considering different $\chi$ and estimated the error in the diffusion constant $D$ by fitting the prediction in Eq. (19) to different time windows.

## C    Ruelle-Pollicott resonances through weak dephasing

In this section we demonstrate the weak Lindbladian dissipation-based method, proposed in Ref. [15], on circuits without any conservation, and briefly show it gives the same results as the truncated propagator method. Additionally, we show that in circuits we studied, the convergence of the Lindblad method can sometimes be rather tricky.

The idea is [15] to introduce some kind of Lindbladian dissipation of strength $\gamma$, obtaining the open system version of the Heisenberg propagator $\mathcal{U}_\gamma$ (which, in this case, is

the exponent of the Lindbladian). The leading RP resonances $\lambda$ then correspond to the leading eigenvalues of the Heisenberg propagator $\lambda_{\mathcal{U}_\gamma}$ in the zero noise $\gamma \to 0^+$ limit and in the thermodynamic limit, i.e.,

$$\lambda = \lim_{\gamma \to 0^+} \lim_{N \to \infty} \lambda_{\mathcal{U}_\gamma}, \tag{34}$$

where $N$ is the system size. The order of limits is important, the opposite order always gives 0.

What kind of dissipation to take is not entirely clear, likely some locality is required. What we want to demonstrate is that the speed of convergence of the above limits might be strongly dependent on the chosen form, or, equivalently, on the chosen 2-qubit gate $V$ for a given dissipation. In this paper we fix dissipation to dephasing and test circuits with different gates $V$. More specifically, the considered system's Heisenberg propagator is

$$\mathcal{D}_\gamma(A) := \gamma \sum_{j=1}^{N} \left( \sigma_j^z A \sigma_j^z - A \right), \tag{35}$$

$$\mathcal{U}_\gamma(A) := e^{\mathcal{D}_\gamma}(U^\dagger A U),$$

where $U$ is the brickwall circuit Floquet propagator without magnetization conservation defined in Eq. (12). In other words, every time step consists of first evolving with the brickwall circuit propagator and then applying dephasing with strength $\gamma$ on every site.

One can calculate the leading eigenvalue of $\mathcal{U}_\gamma$ with an iterative solver (similar to what we did for the truncated propagator, described in Appendix B.1) for different $N$ and $\gamma$. Extrapolation to zero noise and the thermodynamic limit must be done carefully [15]. Namely, if one observes the curve $|\lambda_{\mathcal{U}_\gamma}|$ for fixed $N$, we see that it "breaks" (i.e., abruptly changes its slope) at some small $\gamma$. This change has a physical origin: for small $\gamma$ the scattering length becomes larger than the system size and one starts to be influenced by finite-size (boundary) effects. Therefore, in order to obtain the correct decay rate of the bulk physics from a finite-$N$ data one has to extrapolate the curve $|\lambda_{\mathcal{U}_\gamma}|$ from finite (large) $\gamma$ down to $\gamma = 0$.

An example of such behavior is seen in numerical data for one realization of a brickwall circuit (with a rather tricky instance of $V$) shown in Fig. 10a. Instead of $\lambda$ we plot $\nu_{\mathcal{U}_\gamma} = -\log\left|\lambda_{\mathcal{U}_\gamma}\right|$, like in Ref. [15]. We observe $\nu_{\mathcal{U}_\gamma}$ is independent of the system size $N$ for big enough $\gamma \gtrsim 0.15$. From data at $\gamma > 0.15$, we can extrapolate (black dashed curve) to obtain the prediction for the leading RP resonance $|\lambda| \approx e^{-0.403} \approx 0.668$. Comparing this to the value obtained by the truncated propagator (Fig. 10b) and to decay of correlation functions (Fig. 10c), we see it is incorrect. The long time decay is actually governed by $|\lambda| \approx e^{-0.181} \approx 0.835$, which the truncated propagator predicts correctly (in this case, this is the maximum $\lambda_1(k)$ over all $k$, which occurs at $k = 0$).

If we study $\nu_{\mathcal{U}_\gamma}$ more carefully though, we can get the correct RP resonance from the Lindbladian approach too. Calculating $\nu_{\mathcal{U}_\gamma}$ for smaller $\gamma$, we in fact observe another breaking of the curve. This time, the curves for different $N$ do not overlap, but we can still extrapolate for each $N$ separately and then try to do $N \to \infty$. Such a procedure is also shown in Fig. 10a (colored dotted curves), with the convergence with $N$ shown in the inset (a.i). While the procedure does not yet converge for accessible $N$, it is in the vicinity of what appears to be the correct RP resonance as obtained from the truncated propagator. Additionally, in Fig. 10 we see that the early time behavior of the magnetization correlation function seems to be governed by $|\lambda| \approx 0.668$. The initial breaking of the $\nu_{\mathcal{U}_\gamma}$ therefore might not be spurious, but rather an indication of a subleading resonance.

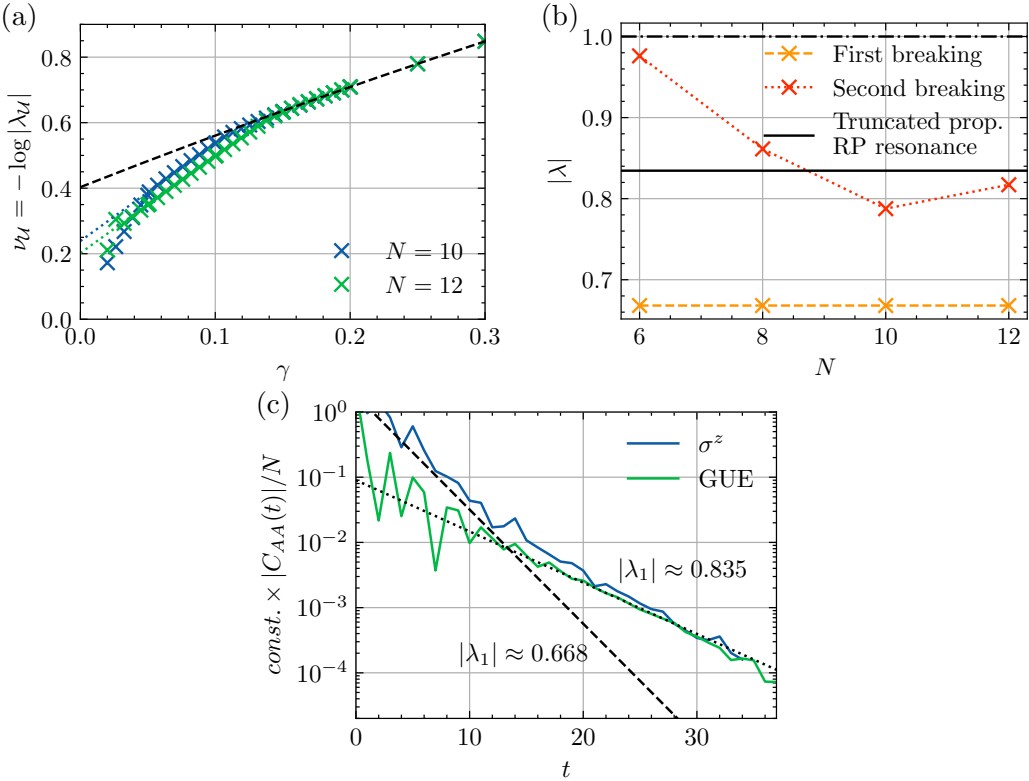

Figure 10: Extraction of the leading RP resonance by a weak-dephasing Lindbladian [see Eq. (35)], an analogous figure to Fig. 2. (a) The leading Liouvillian eigenvalue for the dissipative brickwall quantum circuit with a Haar-random gate $V$. Dashed and dotted lines denote extrapolation to zero noise. (b) The convergence of the second breaking extrapolation [dotted lines in (a)] compared to the first breaking extrapolation [dashed lines (a)] and the RP resonance estimated from $r = 12$ truncated propagator (the biggest gap is at $k = 0$). (c) Infinite temperature autocorrelation function of extensive $k = 0$ observables, defined in Eq. (4). The local densities are either $\sigma^z$ (i.e., $z$ magnetization) or chosen randomly according to the Gaussian unitary ensemble (GUE) [36] with support $r = 2$. The dashed and dotted lines denote the decays $\propto |\lambda|^t$ governed by the first and second breaking predictions, respectively. The numerics are done in a finite circuit with $N = 32$ sites, the $\sigma^z$ correlation function is multiplied by 9 for better presentation.

We must emphasize that the showcased behavior is not generic. Double breaking is not always observed; sometimes the correct RP resonance is given by the first breaking, sometimes not even by the second. Additionally, we do not necessarily see a 2-step relaxation in generic correlation functions. All in all, this example showcases our experience that the Lindbladian method is less robust. Taking the double limit involves making extrapolations that are sometimes hard to perform. Furthermore, if we would want to treat systems with a conservation law, like magnetization, the method is much more cumbersome, and it is not clear how to do the momentum resolution efficiently.

# D    Plateaus in correlation functions due to the powers of the conserved quantity

Powers of a conserved quantity are also conserved quantities. When studying physical properties of many-body systems they can be often ignored because a power of a local operator is a non-local operator. In finite systems of size $N$, however, their impact can be non-negligible. Namely, local observables can have a finite overlap with some power of a conserved quantity and thus cause its correlation function to saturate to a higher value than the expected fluctuations (see Appendix B.2). Although this mechanism is simple, we explicitly demonstrate it in the case of magnetization conservation, since one needs to be aware of it when interpreting finite-size numerics.

Let $Q$ be a conserved quantity, i.e., $\mathcal{U}(Q) = Q$, which means its correlation functions is constant,

$$C_{QQ}(t) = \langle Q(t)Q(0) \rangle = \langle Q, Q \rangle, \tag{36}$$

where the angled brackets denote the Hilbert-Schmidt inner product, $\langle A, B \rangle := \frac{1}{2^N} \operatorname{tr} A^\dagger B$. We additionally assumed $Q$ to be Hermitian and traceless for simplicity, although a similar conclusion can be made also in the general case.

If an observable overlaps with $Q$, $c_Q := \frac{\langle Q, A \rangle}{\langle Q, Q \rangle} \neq 0$, we also expect its correlation function to be constant after a long enough time. Namely, we can decompose $A$ in an orthogonal basis including $Q$, obtaining

$$C_{AA}(t) = \langle A(t)A(0) \rangle = \langle c_Q Q(t) + \cdots, c_Q Q(0) + \cdots \rangle$$

$$\xrightarrow[t \to \infty]{} |c_Q|^2 \langle Q, Q \rangle = \frac{\langle Q, A \rangle^2}{\langle Q, Q \rangle}. \tag{37}$$

Here dots denote the decaying part, i.e, we assumed the system to be mixing and that $Q$ is the only conserved quantity $A$ overlaps with. We also assumed $A$ to be traceless for simplicity.

Let's now demonstrate this with the correlation function of a 2-site extensive observable

$$A = \sum_j \sigma_j^z \sigma_{j+1}^z \tag{38}$$

evolved by a magnetization-conserving circuit. $A$ has zero overlap with the conserved $M = \sum_j \sigma_j^z$, so its correlation function should decay to 0 in the thermodynamic limit $N \to \infty$. While this is true, it however has a non-zero overlap with powers of $M$ and its correlation function will therefore exhibit a finite-size plateau.

Although $A$ overlaps with all even powers of magnetization, the leading order of the plateau in $1/N$ will be caused by $M^2$. For $Q$ one takes the traceless part of $M^2$, $Q = M^2 - N\mathbb{1}$. A short calculation gives

$$\langle M^2 - N\mathbb{1}, A \rangle = 2N, \tag{39}$$

$$\langle M^2 - N\mathbb{1}, M^2 - N\mathbb{1} \rangle = 2N(N-1). \tag{40}$$

Which results in the finite-size plateau

$$\frac{1}{N} C_{AA}(t \to \infty) = \frac{1}{N} \frac{\langle Q, A \rangle^2}{\langle Q, Q \rangle} = \frac{2}{N-1}. \tag{41}$$

We show the result for the correlation function normalized by $1/N$, since for extensive observables $C_{AA}(0) \propto N$.

The correlation functions of $A$ in a homogeneous circuit with a random 3-site gate (i.e., the one described in Sec. 3) and different $N$ are shown in Fig. 11a. The plateaus are shown in Fig. 11b and match the predicted $\sim 1/N$ scaling at large $N$.

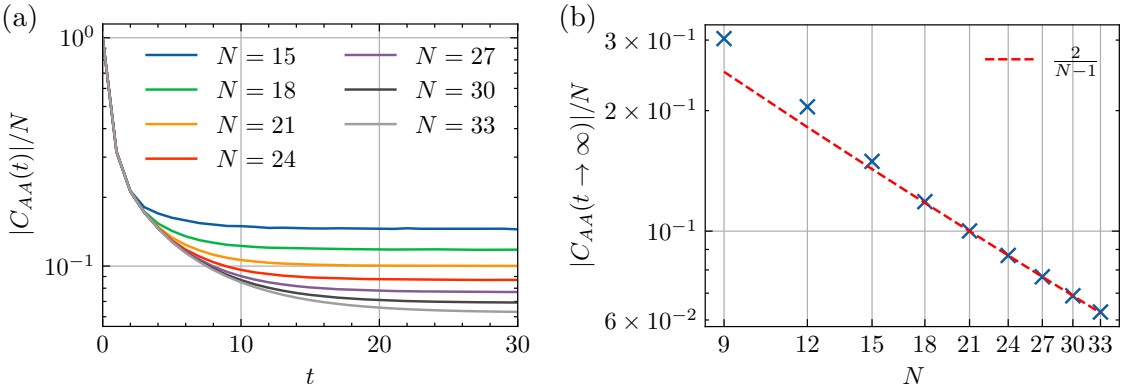

Figure 11: Finite-size plateaus in the autocorrelation function of $A = \sum_j \sigma^z_j \sigma^z_{j+1}$. Shown are numerical results for a circuit with a random 3-site magnetization-conserving gate (Sec. 3) and different $N$. Subfigure (a) shows time dependence, while (b) shows the values of plateaus in (a). The red dashed line are the leading asymptotics (41).

## E    Stretched-exponential decay of local correlation functions

Recently [49], it was argued that all local correlation functions unrelated to transport decay as stretched exponentials in diffusive one-dimensional systems, that is as $\sim \mathrm{e}^{-Bt^\alpha}$ for some $B$ and $\alpha$. For translationally-invariant Floquet systems, Ref. [49] predicts $\alpha = 2/3$ and numerically confirms the prediction for magnetization-conserving brickwall circuits.

    We check this in the 3-site generalized brickwall circuit (see Sec. 3) by analyzing the local correlation function of $\sigma^x$ magnetization, which is not expected to be described by an effective (hydrodynamic) theory of transport [46, 49]. The numerics are shown in Fig. 12, and we average over space to obtain better statistics. The average is done on the absolute value squared, since averaging just $\langle \sigma^x(i,t)\sigma^x(0,0) \rangle$ would yield the extensive $\sigma^x$ magnetization correlation function. While the numerics shown suggest the decay is better described by a stretched exponential with $\alpha = 2/3$ than an exponential, fitting the power $\alpha$ has a big uncertainty (we obtain $\alpha \sim 0.4 - 0.8$ for different models with fitting uncertainties of the order of $\sim 0.1$).

    As argued in Sec. 4.2, stretched-exponential decays are governed by RP continuums with closing gap. An interesting open question is what happens for correlation functions of extensive observables, which are essentially the Fourier transform of local observables. Our numerics seem to indicate exponential decay, at least for big $k$ (see Sec. 3.2). Asymptotics for correlation functions for $k = 0$ are less clear numerically, the extensive $x$ magnetization does not seem to be in the asymptotic regime for accessible time scales and additionally heavily oscillates.

## F    Local spin current

In this section, we provide additional details about the diffusive behavior of the local spin current discussed in Sec. 4.2.1. The spin current $j$ is defined by the discrete continuity equation, Eq. (21). The diffusive predictions come from the diffusive local correlation function of magnetization, Eq. (15), and the continuity equation. For clarity, we use the

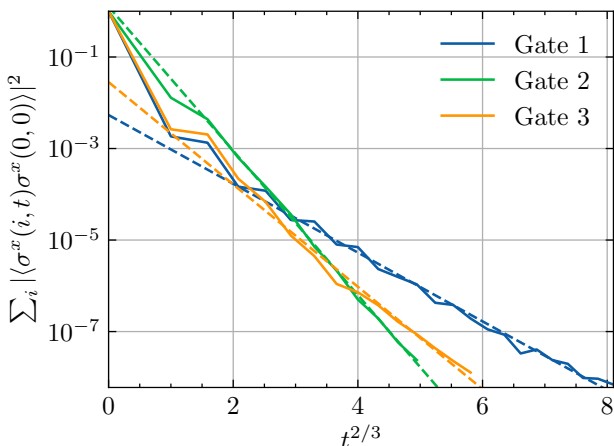

Figure 12: Local correlation function of $\sigma^x$ magnetization. The correlation function is calculated in a circuit with a random 3-site magnetization-conserving gate (Sec. 3) with $N = 33$ sites and is averaged over space. Dashed lines show a stretched exponential $e^{-Bt^{2/3}}$ fit.

continuous space-time continuity equation

$$\partial_t \sigma^z(x,t) = -\partial_x j(x,t), \tag{42}$$

where $j(x,t)$ is the continuous version of the spin current from Eq. (21). We derive

$$
\begin{aligned}
\langle j(x,t)j(0,0)\rangle &= -\int_{-\infty}^{x} \mathrm{d}x' \partial_t \left\langle \sigma^z(x',t)j(0,0)\right\rangle \\
&= -\int_{-\infty}^{x} \mathrm{d}x' \partial_t \left\langle \sigma^z(0,0)j(-x',-t)\right\rangle \\
&= \int_{-\infty}^{x} \mathrm{d}x' \int_{-\infty}^{x'} \mathrm{d}x'' \partial_t^2 \left\langle \sigma^z(0,0)\sigma^z(-x'',-t)\right\rangle \\
&= \int_{-\infty}^{x} \mathrm{d}x' \int_{-\infty}^{x'} \mathrm{d}x'' \partial_t^2 \left\langle \sigma^z(x'',t)\sigma^z(0,0)\right\rangle,
\end{aligned}
\tag{43}
$$

where we used translational invariance and the cyclic property of trace. Plugging in the diffusive magnetization correlation function (15), we obtain

$$\langle j(x,t)j(0,0)\rangle = \frac{x^2 - 2Dt}{8\sqrt{\pi D}t^{5/2}} e^{-\frac{x^2}{4Dt}}. \tag{44}$$

Similarly, as in Sec. 3.1, the correlation function of extensive current with good quasi-momentum $J_k := \sum_l e^{-ilk/3} j_l$ can now be studied. We obtain

$$
\begin{aligned}
\frac{1}{N}\left\langle J_k(t)J_k(0)^\dagger \right\rangle &\to \int \langle j(x,t)j(0,0)\rangle\, e^{-ikx}\, \mathrm{d}x \\
&\propto k^2 e^{-Dk^2 t}.
\end{aligned}
\tag{45}
$$

The decay is, therefore, governed by an RP resonance with the same $k$ dependence as the one governing the decay of magnetization [cf. Eq. (16)], with the prefactor $k^2$ interpreted as an expansion coefficient.

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
