# Peer review of "Ruelle-Pollicott resonances of diffusive U(1)-invariant qubit circuits"

_SciPost Physics_

## Round 1 · Referee Report · Ilya Vilkoviskiy (Referee 1) · 2025-8-19

The referee discloses that the following generative AI tools have been used in the preparation of this report:
Generative AI tools were used solely to improve the English and clarity of the report. All intellectual content of the report is entirely human-generated.
Strengths
-
Novel extension to conserved charges: The authors present a new method for extending the Ruelle–Pollicott resonance framework to systems with conserved charges.
-
Clarity of presentation: The paper is written in a clear and accessible manner, making complex concepts understandable to a broad audience.
-
Benchmarking against existing methods: The authors compare their approach with established techniques, such as truncation via an additional Lindbladian dissipation and direct Time-Evolving Block Decimation (TEBD) evolution, carefully discussing the nuances, strengths, and weaknesses of each.
-
Complementary momentum-space perspective: While the method does not yet show a clear advantage over state-of-the-art approaches, it offers a complementary view by enabling direct examination of correlation functions in momentum space—something not as straightforward in existing algorithms (e.g., TEBD).
-
In-depth spectral analysis: The authors provide a detailed discussion on the possible existence of subleading isolated RP resonances and on corrections to the asymptotic decay of correlation functions.
-
General applicability to strongly interacting systems: The framework offers a generic tool for studying strongly interacting Floquet circuits, opening the way to future generalizations for Hamiltonian systems and different symmetry classes.
Weaknesses
-
No clear improvement in transport coefficient extraction: The proposed method does not demonstrate a clear advantage in determining transport dynamical exponents or the diffusion constant, producing results comparable to those from existing techniques.
-
Limited new physical insights: Although the approach is novel and original, it does not substantially advance our understanding of chaotic systems with U(1) conservation laws.
Report
The paper clearly meets two of the four specific SciPost criteria:
-
Providing a novel and synergetic link between different research areas.
-
Opening a new pathway in an existing or entirely new research direction, with clear potential for multi-pronged follow-up work.
It also clearly satisfies the general acceptance criteria.
I therefore recommend the paper for publication.
Requested changes
As a few minor suggestions:
-
The naming “cases” might be misleading, as it could be interpreted as referring to different models or physical regimes. Since these instead correspond to different samples of Haar-random unitaries, it may be clearer to rename them as “instances” or “samples”.
-
The reference to k≃2 at the top of page 10 is potentially confusing, as in the corresponding graph k is expressed in units of π. Referring instead to k≃0.6π would make it consistent.
Recommendation
Publish (easily meets expectations and criteria for this Journal; among top 50%)
We thank the referee for their positive comments and careful reading. We have incorporated both of the suggested changes in the revised version.

Author: Urban Duh on 2025-10-02 [id 5880]
(in reply to Report 3 on 2025-09-17)We thank the referee for the positive report and comments that helped us improve the manuscript. We address their points below:
Question 1:
I was a bit confused by the proper order of limits in Eq. (11). In a finite system, taking $r \to \infty$ should restore unitarity, so that all eigenvalues have unit modulus. I'm assuming the authors imagine an infinite system, but in that case, the spectrum might be continuous (as they indeed argue happens in the systems they study) so that the approximation using only the leading eigenvalue need not be valid. Could they comment on this issue?
Answer 1:
The referee's assumption is correct; we had an infinite system in mind, which we additionally clarified in the revised manuscript. It is also true that in the case of a continuous spectrum, the approximation using only the leading eigenvalue might not be correct (one of the simplest examples is precisely the power-law decay of the local magnetization correlation function we discuss in the paper). The purpose of the paragraph in question is to provide the reader with an intuition for the case without symmetries, where a single RP resonance is well-defined and, for this reason, we postpone the discussions about continuous spectra to Sec. 4.2. To avoid confusion, we emphasize in the revised version that we have in mind a situation with an isolated $\lambda_1$ and finite expansion coefficient, i.e., not infinitesimally small in the continuous limit, like the expansion coefficient $c(\nu = 0)$ in footnote 6 (previously footnote 5).
Question 2:
In Eq. (19), in the second formula, is there a subtraction of the t=0 value missing?
Answer 2:
We thank the referee for spotting this error, we have corrected it.
Question 3:
In Section 4.1, the authors give a heuristic argument for the size of the subleading RP resonances coming from powers of the conserved charge. Could this be turned into a rigorous bound, using the locality of the circuit?
Answer 3:
A rigorous bound can be made, see the new footnote 5. However, it contains an a priori unknown condition number. Because a potential better bound on the subleading eigenvalues due to powers of the conserved charges does not have any immediate physical significance -- if anything one would want to get rid of them in order to have better resolution of nontrivial eigenvalues -- we did not pursue them any further beyond the simple estimate given in Eq. (20).
Question 4:
How is Eq. (24) consistent with diffusion (since the diffusion constant itself is related to the current-current autocorrelation)? Is this an issue of the proper order of limits?
Answer 4:
We thank the referee for this remark. Eq. (24), as well as (22) and (15) from which it is derived, are expected to hold only for long times. We have emphasized this in the revised manuscript, adding also the new footnote 9. The main message of Eq. (24) is that at long times, $C_{JJ}$ is controlled by the subleading behavior of (or, in other words, by the corrections to) the local magnetization correlation function. The Green-Kubo formula for the diffusion constant (which the referee probably had in mind) is, on the other hand, dominated by short times, since the integral of $C_{JJ}$ has the largest contributions at short times (crucially, the connection between short and long times can be made through linear response). Therefore, Eq. (24) does not contradict it.
Question 5:
The authors mention that Ref. 47 predicted a power-law decay for current-current correlations. What is the power and how does it relate to their numerical results in Fig. 9?
Answer 5:
Ref. 47 predicts the local magnetization correlation function to be an infinite sum of power-law decaying terms, see their Eq. (A1). However, many of the terms give a zero contribution to $C_{JJ}$ for the same reason as the leading order diffusion prediction: the corresponding local current correlation function is a total derivative in $x$ [see our Eq. (22)]. In fact, all the terms explicitly evaluated in Ref. 47 give a 0 contribution, which is why it isn't obvious what the asymptotic decay of $C_{JJ}$ should be. Additionally, the terms introduce new model-dependent constants which can be 0 for the model in question. We have added additional clarification about this at the end of Sec. 4.2.1, but the main message remains: the asymptotics of $C_{JJ}$ are still not completely understood.
Question 6:
The RP resonance approach to hydrodynamics appears similar to ideas that appeared under other names in recent years, particularly PRL 131, 220403 by Ogunnaike et al and PRX Quantum 5, 040330 by Moudgalya and Motrunich. I think these papers should be mentioned somewhere in the manuscript.
Answer 6:
Thank you, we were not aware of those works. While they do study diffusion, the setting they consider and where their methods apply is considerably different from what we study. In both papers one has an exact description (of the average dynamics) in terms of a dissipative propagator. In such cases when one starts with a dissipative description things are simple; one can get the relaxation rate from the gap (this connection is well known and has been studied e.g. in the Lindblad and even before in the Markov chain community). The RP idea is very different -- starting with an exact unitary propagator (no noise or time dependence) the question is how to get the effective relaxation. The method used in the mentioned papers can be applied for instance to time-dependent (Brownian) circuits or directly to systems with an explicit dissipation. Our system is, on the other hand, deterministic and time-independent. There are many papers on similar random (time-dependent) systems, as well as on exact solutions (including author's own papers), but they are only very weakly related to what we are doing. Therefore, we have decided not to cite any of those. Singling out 1 or 2 papers out of many would be odd, and we want to avoid grouping citations like [10-30] as it does not convey any useful information.
Question 7:
A few typos: "Infinitely dimensional", "unit call" and "does not conserve only"
Answer 7:
We thank the referee for their careful reading, the typos have been corrected.

---

## Round 1 · Referee Report · Pieter W. Claeys (Referee 2) · 2025-9-1

Strengths
1- Clear connection between Ruelle-Pollicott resonances and hydrodynamics. 2- The presented results are expected to be applicable to a wide range of U(1)-invariant circuits.
Weaknesses
Report
After introducing RP resonances and the truncated quasi-momentum-dependent propagator, with truncation based on operator support, the authors study the quasi-momentum dependence of the leading RP resonance. It is shown that this eigenvalue has a Gaussian dependence near zero quasi-momentum, as expected from hydrodynamics, from which the diffusion constant can be extracted. For larger quasi-momentum this scaling is lost, resulting in generic exponential decay. It is additionally shown how higher power of the conserved charge govern subleading eigenvalues, and conjectured that hydrodynamic tails give rise to a continuum of RP resonances.
The paper is convincing and clearly written, with illustrative figures and numerics, and the results (while perhaps unsurprising) are sure to be of interest to the community studying quantum dynamics. It easily meets the SciPost Physics criteria of providing a "novel and synergetic link between different research areas" and using this RP approach to calculate diffusion constants and study anomalous hydrodynamics presents "clear potential for multi-pronged follow-up work". For these reasons I am happy to recommend this paper for publication in SciPost Physics provided some minor comments are addressed.
Requested changes
1- The study of transport by truncating the dynamics to operators with a finite local support size is closely related to the dissipation-assisted operator evolution (DAOE) approach to quantum hydrodynamics [see, e.g. Phys. Rev. B 105, 075131 (2022) and related works]. I would have expected to see these works referenced, and I think the current manuscript would be strengthened by discussing this connection and possibly outlining how the DAOE truncation (based on operator length, i.e. number of nontrivial Pauli operators) would modify the conclusions.
2- In the summary of results and in Section 2.3, it is mentioned that the quasi-momentum dependence of the leading PR resonance is in general non-trivial, even in systems without conservation laws. Here it would be useful to discuss the connection with Ref. [22], one of the authors' previous works.
3- I'm not sure how meaningful it is to include data that is not yet converged without showing any indicator of convergence, as in Fig. 6. It is mentioned in the text that many of the presented eigenvalues appear to be converging to 1, but it would be useful to show some data to support this claim.
4-It's not obvious to me how Eq. (20) follows from the above argument. Could the authors clarify? From Fig. 7 it's also not clear to me that this argument should return the correct scaling for higher orders of M.
Recommendation
Publish (easily meets expectations and criteria for this Journal; among top 50%)
We thank the referee for the positive report and comments that helped us improve the manuscript. We address their points below:
Question 1:
The study of transport by truncating the dynamics to operators with a finite local support size is closely related to the dissipation-assisted operator evolution (DAOE) approach to quantum hydrodynamics [see, e.g. Phys. Rev. B 105, 075131 (2022) and related works]. I would have expected to see these works referenced, and I think the current manuscript would be strengthened by discussing this connection and possibly outlining how the DAOE truncation (based on operator length, i.e. number of nontrivial Pauli operators) would modify the conclusions.
Answer 1:
We thank the referee for reminding us about the DAOE papers, which definitely deserved to be mentioned due to a somewhat similar idea (mentioned now in Sec. 1.2 and Conclusion). The differences are that DAOE's truncation is based on the operator length rather than support, the absence of a sharp cutoff, and on a technical level, that we work directly in the infinite system size limit. While a detailed comparison would be interesting, it is out of the scope of our paper. We can offer some results in Ref. [34] that suggest our approach might be better in the considered cases (see also new footnote 10).
Question 2:
In the summary of results and in Section 2.3, it is mentioned that the quasi-momentum dependence of the leading PR resonance is in general non-trivial, even in systems without conservation laws. Here it would be useful to discuss the connection with Ref. [22], one of the authors' previous works.
Answer 2:
At present we do not have any detailed understanding of the behavior at large $k$, similarly as also in Ref. [22]. In the revised manuscript we have added a comment that our observations are similar to the those in the existing literature.
Question 3:
I'm not sure how meaningful it is to include data that is not yet converged without showing any indicator of convergence, as in Fig. 6. It is mentioned in the text that many of the presented eigenvalues appear to be converging to 1, but it would be useful to show some data to support this claim.
Answer 3:
At $k=0$, the approach of subleading eigenvalues towards $1$ can be seen in Fig. 7. Details about that are given in Sec. 4.1, we have additionally emphasized this in the figure's caption and in the main text.
Question 4:
It's not obvious to me how Eq. (20) follows from the above argument. Could the authors clarify? From Fig. 7 it's also not clear to me that this argument should return the correct scaling for higher orders of M.
Answer 4:
We rewrote two paragraphs preceding Eq. (20) and added a new footnote 5 that clarifies the heuristic estimate leading to Eq. (20). Note that the argument is non-rigorous, essentially just a term counting, numerically, though, it seems that it gives the correct $1/r$ scaling of the gap.

---

## Round 2 · Author Response

List of changes
- Changes in the text suggested by referees, marked in red
- A number of minor changes in the text to improve readability and correct typos (not marked in red)
- Improved numerics in Figures 2c, 4, 6, 9, 10c, 11, 12 (reflected in captions and/or text in figures and marked in red); there are no changes in conclusions based on numerics

---

## Round 2 · List of Changes

- Changes in the text suggested by referees, marked in red
- A number of minor changes in the text to improve readability and correct typos (not marked in red)
- Improved numerics in Figures 2c, 4, 6, 9, 10c, 11, 12 (reflected in captions and/or text in figures and marked in red); there are no changes in conclusions based on numerics

---

## Editorial Decision

unknown